# SoFlow: Solution Flow Models for One-Step Generative Modeling

**Tianze Luo, Haotian Yuan, Zhuang Liu**
Princeton University

## Abstract

The multi-step denoising process in diffusion and Flow Matching models causes major efficiency issues, which motivates research on few-step generation. We present Solution Flow Models (SoFlow), a framework for one-step generation from scratch. By analyzing the relationship between the velocity function and the solution function of the velocity ordinary differential equation (ODE), we propose a Flow Matching loss and a solution consistency loss to train our models. The Flow Matching loss allows our models to provide estimated velocity fields for Classifier-Free Guidance (CFG) during training, which improves generation performance. Notably, our consistency loss does not require the calculation of the Jacobian-vector product (JVP), a common requirement in recent works that is not well-optimized in deep learning frameworks like PyTorch. Experimental results indicate that, when trained from scratch using the same Diffusion Transformer (DiT) architecture and an equal number of training epochs, our models achieve better FID-50K scores than MeanFlow models on the ImageNet 256×256 dataset. Our code is available at `github.com/zlab-princeton/SoFlow`.

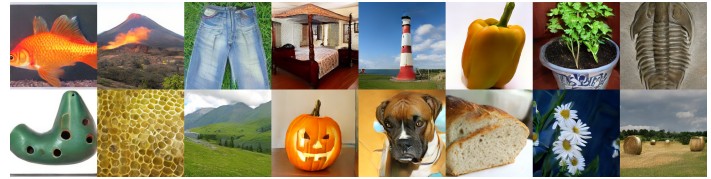

| Size | MeanFlow | SoFlow |
|------|----------|--------|
| B/2  | 6.17     | 4.85   |
| M/2  | 5.01     | 3.73   |
| L/2  | 3.84     | 3.20   |
| XL/2 | 3.43     | **2.96** |

(a) 1-NFE samples from our XL/2 model      (b) 1-NFE FID-50K comparison

Figure 1: **Visual samples and quantitative comparison.** (a) One-step samples generated by our Solution Flow Models on the ImageNet 256×256 dataset. (b) With the same Diffusion Transformer (Peebles & Xie, 2023) architecture and the same number of training epochs, our models (SoFlow) consistently achieve better 1-NFE FID-50K scores than MeanFlow (Geng et al., 2025) models on the ImageNet 256×256 dataset.

## 1 Introduction

Diffusion models (Sohl-Dickstein et al., 2015; Song & Ermon, 2019; Ho et al., 2020; Song et al., 2020) and Flow Matching models (Lipman et al., 2022; Liu et al., 2022; Albergo & Vanden-Eijnden, 2022) have emerged as foundational frameworks in generative modeling. Diffusion models operate by systematically adding noise to data and then learning a denoising process to generate high-quality samples. Flow Matching offers a more direct alternative, modeling the velocity fields that transport a simple prior distribution to a complex data distribution. Despite their power and success across various generative tasks (Esser et al., 2024; Ma et al., 2024; Polyak et al., 2024), both approaches rely on an iterative, multi-step sampling process, which hinders their generation efficiency.

Addressing this latency is becoming a key area of research. Consistency Models (Song et al., 2023; Song & Dhariwal, 2023; Geng et al., 2024; Lu & Song, 2024) and related techniques (Kim et al., 2023; Wang et al., 2025a; Frans et al., 2024; Heek et al., 2024) have gained prominence by enabling rapid, few-step generation. These methods learn a direct mapping from any point on a generative trajectory to a consistent, "clean" output, bypassing the need for iterative refinement. However, this paradigm introduces significant challenges. Consistency models trained from scratch often fail to

leverage Classifier-Free Guidance (CFG) for enhancing sample quality, and they are further hindered by instability caused by changing optimization targets. Recent works (Peng et al., 2025; Geng et al., 2025) address this instability by incorporating a Flow Matching loss. However, this approach introduces a new computational bottleneck: it relies on Jacobian-vector product (JVP) calculations, which are much less optimized than forward propagation in frameworks such as PyTorch.

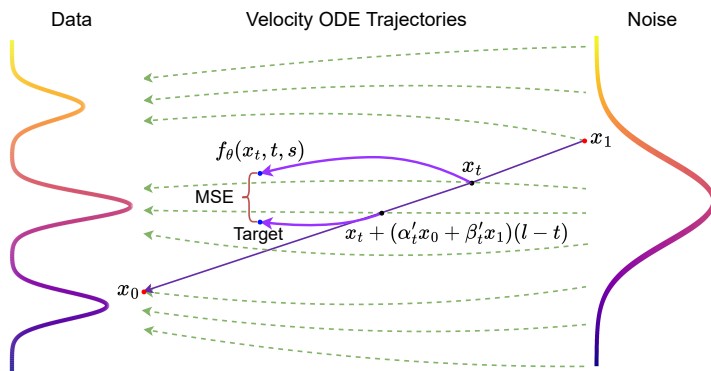

Figure 2: **Illustration of solution consistency loss.** The plot shows a straight-line Flow Matching trajectory defined by a data-noise pair $(x_0, x_1)$, where $x_t$ is the intermediate point given by $x_t = \alpha_t x_0 + \beta_t x_1$ ($\alpha_t, \beta_t$ are $C^1$ functions with $\alpha_0 = \beta_1 = 1, \alpha_1 = \beta_0 = 0$). Given three time points $s < l < t$, the mean squared error is computed between our model $f_\theta(x_t, t, s)$ and the stop-gradient target $f_{\theta^-}(x_t + (\alpha'_t x_0 + \beta'_t x_1)(l - t), l, s)$.

In this work, we introduce a new approach for one-step generation that avoids these limitations. Instead of relying on iterative ODE solvers, we propose directly learning the solution function of the velocity ODE defined by Flow Matching to generate high-quality samples, as shown in Figure 1a, Figure 4, and Figure 5. We denote this function as $f(x_t, t, s)$, which explicitly maps a state $x_t$ at time $t$ to its evolved state $x_s$ at another time $s$. To learn it with a parameterized model $f_\theta(x_t, t, s)$, we first analyze the properties that enable a neural network to serve as a valid solution function.

Based on this analysis, we formulate a training objective comprising a Flow Matching loss and a solution consistency loss (see Figure 2). The resulting model, $f_\theta(x_t, t, s)$, not only accommodates the Flow Matching objective and CFG naturally but also eliminates the need for expensive JVP calculations during training. Our experimental results demonstrate the effectiveness of this approach, showing that our models achieve superior FID-50K scores compared to MeanFlow models on the ImageNet 256×256 dataset, using the same Diffusion Transformer (DiT) architecture and the same number of training epochs.

## 2  RELATED WORK

**Diffusion, Flow Matching, and Stochastic Interpolants.** Diffusion models (Sohl-Dickstein et al., 2015; Song et al., 2020; Kingma et al., 2021; Karras et al., 2022) and Flow Matching (Lipman et al., 2022; Liu et al., 2022) are widely adopted generative modeling frameworks. These approaches either progressively corrupt data with noise and train a neural network to denoise it, or learn a velocity field that governs a transformation from the data distribution to a simple prior. They have been scaled successfully for image generation (Rombach et al., 2022; Saharia et al., 2022; Podell et al., 2023) and video generation (Ho et al., 2022; Brooks et al., 2024) tasks. Stochastic interpolants (Albergo & Vanden-Eijnden, 2022; Albergo et al., 2023) build upon these concepts by explicitly defining stochastic trajectories between the data and prior distributions and aligning their associated velocity fields to enable effective distributional transport.

**Few-step Generative Models.** Reducing the number of sampling steps has become an active research direction, driven by the need for faster generation and better theoretical insights. Prior efforts fall into two main streams: (i) distillation-based approaches that compress pre-trained multi-step

models into few-step generators (Salimans & Ho, 2022; Sauer et al., 2024; Geng et al., 2023; Luo et al., 2023b; Yin et al., 2024; Zhou et al., 2024); and (ii) from-scratch training, which learns fast samplers without teachers, most prominently through the consistency-training family (CMs, iCT, ECT, sCT) (Song et al., 2023; Song & Dhariwal, 2023; Geng et al., 2024; Lu & Song, 2024).

Consistency Models (CMs) collapse noised inputs directly to clean data, enabling one- or few-step generation (Song et al., 2023). While first applied mainly in distillation (Luo et al., 2023a; Geng et al., 2024), later works established that they can also be trained from scratch via consistency training (Song & Dhariwal, 2023). Building on this, iCT (Song & Dhariwal, 2023) simplifies objectives and improves stability with robust losses and teacher-free training, while ECT (Geng et al., 2024) introduces a continuous-time ODE formulation that unifies CMs and diffusion. The recent sCT framework (Lu & Song, 2024) further stabilizes continuous-time consistency training, scaling CMs up to billion-parameter regimes. Beyond fixed start and end settings, newer approaches extend CMs to arbitrary timestep transitions, aligning better with continuous-time dynamics.

Specifically, Shortcut Models (Frans et al., 2024) condition on noise level and step size to flexibly support both one- and few-step sampling with shared weights. MeanFlow (Geng et al., 2025) introduces interval-averaged velocities and analyzes the relationship between averaged and instantaneous velocities. IMM (Zhou et al., 2025) matches moments across transitions in a single-stage objective, reducing variance and avoiding multi-stage distillation. Flow-Anchored CMs (Peng et al., 2025) regularize shortcut learning with a Flow Matching anchor, improving stability and generalization. On the distillation side, Align Your Flow (Sabour et al., 2025) unifies CM and FM into continuous-time flow maps effective across arbitrary step counts, further enhanced with autoguidance and adversarial fine-tuning. Finally, Transition Models (Wang et al., 2025b) reformulate generation around exact finite-interval dynamics, unifying few- and many-step regimes and achieving monotonic quality gains as step budgets increase.

## 3 PRELIMINARY: FLOW MATCHING

We first introduce the setting of Flow Matching. Flow Matching models learn a velocity field that transforms a known prior distribution like the standard Gaussian distribution to the data distribution. More precisely, we denote the data distribution as $p(x_0)$, and the prior distribution as $p(x_1)$, which is the standard Gaussian distribution $\mathcal{N}(0, I_n)$ in our setting, where $n$ represents the data dimension.

A general noising process is defined as $x_t = \alpha_t x_0 + \beta_t x_1$, where $x_t \in \mathbb{R}^n, t \in [0, 1]$, $\alpha_t$ and $\beta_t$ are continuously differentiable functions satisfying the boundary conditions $\alpha_0 = 1$, $\alpha_1 = 0$, $\beta_0 = 0$, and $\beta_1 = 1$. Then, the marginal velocity field associated with the noising process is defined as

$$v(x_t, t) = \mathbb{E}_{p(x_0, x_1 | x_t)}[\alpha'_t x_0 + \beta'_t x_1], \qquad (1)$$

which is a conditional expectation given $x_t$ of the conditional velocity field. $\alpha'_t$ and $\beta'_t$ denote the derivatives of $\alpha_t$ and $\beta_t$. According to the Flow Matching framework, given the marginal velocity field $v(x_t, t)$, new samples can be generated by first sampling $x_1 \sim p(x_1)$, and then solving the initial value problem (IVP) of the following velocity ordinary differential equation (ODE) from $t = 1$ to $t = 0$:

$$\frac{dX(t)}{dt} = v(X(t), t), X(1) = x_1. \qquad (2)$$

To construct a generative model based on this principle, a straightforward approach is to approximate the marginal velocity field using a neural network $v_\theta$, parameterized by $\theta$, which is trained by minimizing the following mean squared objective:

$$\mathcal{L}_{\text{Vel}}(\theta) = \mathbb{E}_{t, x_t} \|v_\theta(x_t, t) - v(x_t, t)\|_2^2. \qquad (3)$$

However, directly optimizing a neural network with this loss is impractical since the real velocity field $v(x_t, t)$ is a conditional expectation given $x_t$. To overcome this challenge, a conditional variant of the Flow Matching loss is introduced (Lipman et al., 2022; Liu et al., 2022):

$$\mathcal{L}_{\text{CFM}}(\theta) = \mathbb{E}_{t, x_0, x_1, x_t} \|v_\theta(x_t, t) - (\alpha'_t x_0 + \beta'_t x_1)\|_2^2. \qquad (4)$$

Minimizing $\mathcal{L}_{\text{CFM}}$ is equivalent to minimizing the original objective $\mathcal{L}_{\text{Vel}}$. This loss function is tractable since we only need to sample data-noise pairs to construct targets for the network.

## 4 SOLUTION FLOW MODELS

### 4.1 FORMULATIONS

The main purpose of this work is to investigate how to train a neural network that can directly solve the velocity ODE, thereby eliminating the reliance on numerical ODE solvers in Flow Matching models. We study this problem under the assumption that the velocity field $v(x_t, t) \in \mathbb{R}^n$, with $x_t \in \mathbb{R}^n$ and $t \in [0, 1]$, is continuously differentiable and globally Lipschitz continuous with respect to $x_t$, i.e., $\|v(x_t, t) - v(y_t, t)\|_2 \le L_v \|x_t - y_t\|_2, \forall x_t, y_t \in \mathbb{R}^n, t \in [0, 1]$.

According to the existence and uniqueness theorem of ODEs, these two assumptions guarantee that, given initial condition $X(t) = x_t$, where $x_t \in \mathbb{R}^n, t \in [0, 1]$ can be arbitrarily chosen, a unique solution $X(s), s \in [0, t]$ to the velocity ODE $\frac{dX(s)}{ds} = v(X(s), s)$ exists. Since the initial condition can be varied, we denote this unique solution by notation $f(x_t, t, s)$. Then we immediately have two identities by the definition of the ODE, for any $x_t \in \mathbb{R}^n, 0 \le s \le t \le 1$:

$$f(x_t, t, t) = x_t, \tag{5}$$

$$\partial_3 f(x_t, t, s) = v(f(x_t, t, s), s), \tag{6}$$

where $\partial_3 f(x_t, t, s) \in \mathbb{R}^n$ is the partial derivative function of $f(x_t, t, s)$ with respect to the third variable. Equivalently, the two equations can also be written as the following integral equation:

$$f(x_t, t, s) = x_t + \int_t^s v(f(x_t, t, u), u) du. \tag{7}$$

Since $f(x_t, t, s)$ maps an initial value $x_t$ at time $t$ to the unique solution of the velocity ODE at time $s$, we denote it as the solution function in this paper. Owing to the continuous differentiability of the velocity field $v(x_t, t)$, the solution function $f(x_t, t, s)$ is also continuously differentiable. To realize one-step generation, we need to train a model $f_\theta(x_t, t, s)$ to approximate the ground truth solution function $f(x_t, t, s)$, which is determined uniquely by the velocity field $v(x_t, t)$.

Under the setting discussed above, the following two conditions are sufficient to ensure that $f_\theta(x_t, t, s) = f(x_t, t, s)$ for all $x_t \in \mathbb{R}^n, 0 \le s \le t \le 1$:

$$f_\theta(x_t, t, t) = x_t, \tag{8}$$

$$\partial_1 f_\theta(x_t, t, s) v(x_t, t) + \partial_2 f_\theta(x_t, t, s) = 0. \tag{9}$$

where $\partial_1 f_\theta(x_t, t, s) \in \mathbb{R}^{n \times n}$ is the Jacobian matrix function of $f_\theta(x_t, t, s)$ with respect to the first variable, $v(x_t, t)$ is multiplied with it via a matrix-vector product, and $\partial_2 f_\theta(x_t, t, s) \in \mathbb{R}^n$ is the partial derivative vector function of $f_\theta(x_t, t, s)$ with respect to the second variable.

This can be proven by expanding the following derivative for $0 \le s \le l \le t \le 1$:

$$\frac{\partial}{\partial l}(f_\theta(f(x_t, t, l), l, s)) = \partial_1 f_\theta(f(x_t, t, l), l, s) \underbrace{\partial_3 f(x_t, t, l)}_{=v(f(x_t,t,l),l)} + \partial_2 f_\theta(f(x_t, t, l), l, s) = 0. \tag{10}$$

The expression is equal to 0 due to the conditions in Eq. 6 and Eq. 9. Thus, we know that the model $f_\theta(x_t, t, s)$ is indeed the true solution function $f(x_t, t, s)$ by:

$$f_\theta(x_t, t, s) = f_\theta(f(x_t, t, t), t, s) = f_\theta(f(x_t, t, s), s, s) = f(x_t, t, s), \tag{11}$$

where we use the property that $f(x_t, t, t) = f_\theta(x_t, t, t) = x_t$ and $f_\theta(f(x_t, t, l), l, s)$ is invariant with respect to $l$, allowing us to set $l = t$ and $l = s$ to finish the proof.

### 4.2 LEARNING OBJECTIVES

We now consider how to construct efficient objectives for neural networks to learn the ground truth solution function, according to the two conditions mentioned in the previous section. To ensure the first boundary condition in Eq. 8 is satisfied, we adopt the following parameterization:

$$f_\theta(x_t, t, s) = a(t, s) x_t + b(t, s) F_\theta(x_t, t, s), \tag{12}$$

where $a(t, s)$ and $b(t, s)$ are continuously differentiable scalar functions satisfying $a(t, t) = 1$ and $b(t, t) = 0$ for all $t \in [0, 1]$, and $F_\theta(x_t, t, s)$ denotes a raw neural network. Following the notation

in the last section, we use $\partial_1 a(t, s), \partial_2 a(t, s), \partial_1 b(t, s), \partial_2 b(t, s)$ to denote the partial derivatives of $a(t, s)$ and $b(t, s)$ with respect to the first and second variables. In our experiments, we choose two specific parameterizations, including the Euler parameterization and the trigonometric parameterization:

$$f_\theta(x_t, t, s) = x_t + (s - t)F_\theta(x_t, t, s), \tag{13}$$

$$f_\theta(x_t, t, s) = \cos\left(\frac{\pi}{2}(s - t)\right) x_t + \sin\left(\frac{\pi}{2}(s - t)\right) F_\theta(x_t, t, s). \tag{14}$$

Trivially, these two parameterizations satisfy the boundary condition $f_\theta(x_t, t, t) = x_t$. Then, we construct two loss functions for training using Eq. 9.

**Flow Matching Loss.** Firstly, we consider a special case of Eq. 9 when $t = s$. Recall that the boundary condition Eq. 8 gives $f_\theta(x_t, t, t) = x_t, \forall x_t \in \mathbb{R}^n, t \in [0, 1]$. Thus, we have:

$$\partial_1 f_\theta(x_t, t, t) = I_n, 0 = \left.\frac{\partial f_\theta(x_t, l, l)}{\partial l}\right|_{l=t} = \partial_2 f_\theta(x_t, t, t) + \partial_3 f_\theta(x_t, t, t), \tag{15}$$

where $I_n \in \mathbb{R}^{n \times n}$ is the identity matrix. Then Eq. 9 can be simplified to $v(x_t, t) = \partial_3 f_\theta(x_t, t, t)$. Under our parameterization mentioned above, we have

$$v(x_t, t) = \partial_3 f_\theta(x_t, t, t) = \partial_2 a(t, t)x_t + \partial_2 b(t, t)F_\theta(x_t, t, t) + \underbrace{b(t, t)}_{0} \partial_3 F_\theta(x_t, t, t). \tag{16}$$

where the complex term containing $\partial_3 F_\theta(x_t, t, t)$ is canceled since $b(t, t) = 0$ by our choice of parameterization. Now we obtain a Flow Matching loss for our neural networks:

$$\mathcal{L}_{\text{FM}}(\theta) = \mathbb{E}_{t, x_0, x_1, x_t}\left[\frac{w_{\text{FM}}(t, \text{MSE})}{n} \|\partial_2 a(t, t)x_t + \partial_2 b(t, t)F_\theta(x_t, t, t) - (\alpha_t' x_0 + \beta_t' x_1)\|_2^2\right], \tag{17}$$

where $n$ is the data dimension, $\alpha_t' x_0 + \beta_t' x_1$ is used to replace the intractable marginal velocity field $v(x_t, t) = \mathbb{E}_{p(x_0, x_1 | x_t)}[\alpha_t' x_0 + \beta_t' x_1]$ during the training process following the standard Flow Matching framework. In addition, this velocity term can also be provided by a teacher model in distillation situations, but in this paper, we focus on the from-scratch training setting.

Previous works (Geng et al., 2024; 2025) have demonstrated that choosing an adaptive weighting function is beneficial for few-step generative models. Following their approach, we choose

$$w_{\text{FM}}(t, \text{MSE}) = \frac{1}{|\partial_2 b(t, t)|(\text{MSE} + \epsilon)^p} \tag{18}$$

as our weighting function, where $|\partial_2 b(t, t)|$ is used to balance the raw network's gradients across time, and MSE represents the original mean squared error. Here $\epsilon$ is a smoothing factor to prevent excessively small values, and $p$ is a factor that determines how robust the loss is. For $p = 0$, the objective degenerates to the mean squared error. For $p > 0$, this factor will reduce the contribution of the data points with large errors in a data batch to make the objective more robust.

The original Flow Matching framework samples $t$ uniformly, while more recent works (Esser et al., 2024; Geng et al., 2025) suggest sampling $t$ from a logit-normal distribution. We also sample $t$ from $\sigma(\mathcal{N}(\mu_{\text{FM}}, \sigma_{\text{FM}}^2))$, where $\sigma(\cdot)$ and $\mathcal{N}$ represent the sigmoid function and normal distribution.

**Solution Consistency Loss.** We now consider how to build a training target for the $s < t$ situation using Eq. 9. According to the Taylor expansion, we have an approximation equation:

$$\frac{f_\theta(x_t, t, s) - f_\theta(x_t + v(x_t, t)(l - t), l, s)}{t - l} = (\partial_1 f_\theta(x_t, t, s)v(x_t, t) + \partial_2 f_\theta(x_t, t, s)) + o(1), \tag{19}$$

where $l \in (s, t)$ is close to $t$. We therefore propose a solution consistency loss $\mathcal{L}_{\text{SCM}}(\theta)$:

$$\mathcal{L}_{\text{SCM}}(\theta) = \mathbb{E}_{\substack{t, l, s, \\ x_0, x_1, x_t}}\left[\frac{w_{\text{SCM}}(t, l, s, \text{MSE})}{n} \left\|f_\theta(x_t, t, s) - f_{\theta^-}\left(x_t + (\alpha_t' x_0 + \beta_t' x_1)(l - t), l, s\right)\right\|_2^2\right], \tag{20}$$

where $n$ is the data dimension, $\theta^-$ means applying the stop-gradient operation to the parameters, and $(\alpha'_t x_0 + \beta'_t x_1)$ is again used to replace the intractable $v(x_t, t) = \mathbb{E}_{p(x_0, x_1 | x_t)}[\alpha'_t x_0 + \beta'_t x_1]$ following the common practice. The adaptive weighting is chosen as follows:

$$w_{\text{SCM}}(t, l, s, \text{MSE}) = \frac{1}{(t-l)|b(t,s)|} \times \frac{1}{(\frac{\text{MSE}}{(t-l)^2} + \epsilon)^p}, \tag{21}$$

where the first term ensures the gradient magnitude of the raw network $F_\theta(x_t, t, s)$ is stable, and the second term is again used to provide a more robust loss by scaling down the coefficient for data points with large mean squared errors when $p > 0$. Besides, we divide the mean squared error in the adaptive term by $(t-l)^2$ since its magnitude is proportional to $(t-l)^2$.

As for the sampling method of $t, l, s$ during training, we first sample $t$ and $s$ from two logit-normal distributions, $\sigma(\mathcal{N}(\mu_t, \sigma_t^2))$ and $\sigma(\mathcal{N}(\mu_s, \sigma_s^2))$, respectively. Here, $s$ is clamped to ensure $s < t - 10^{-4}$. We then determine $l$ using the following method:

$$l = t + (s - t) \times r(k, K), \tag{22}$$

where $k, K$ represent the current and total training steps, respectively. The function $r(k, K)$ represents a monotonically decreasing schedule that gradually moves $l$ towards $t$ throughout training. To avoid numerical issues, $l$ is clamped to ensure $l < t - 10^{-4}$. This schedule decreases from an initial value $r_{\text{init}}$ to an end value $r_{\text{end}}$. In our ablation studies, we test exponential, cosine, linear, and constant schedules. For more implementation details, please refer to Appendix A.

The total training loss is a combination of the Flow Matching and solution consistency losses: $L(\theta) = \lambda \mathcal{L}_{\text{FM}}(\theta) + (1 - \lambda)\mathcal{L}_{\text{SCM}}(\theta)$. The parameter $\lambda$ controls the balance between them by determining the fraction of a data batch dedicated to computing $\mathcal{L}_{\text{FM}}(\theta)$, while the remaining fraction is used for $\mathcal{L}_{\text{SCM}}(\theta)$. We perform ablation studies to determine the optimal value of $\lambda$.

### 4.3 CLASSIFIER-FREE GUIDANCE

Classifier-Free Guidance (CFG) (Ho & Salimans, 2022) is a standard technique in diffusion models for enhancing conditional generation. The models are trained with randomly dropped conditions to mix conditional and unconditional data. During inference, CFG is applied by linearly combining predictions from the label-conditional and unconditional models to enhance generation quality.

To apply CFG to our models, we first introduce the ground-truth guided marginal velocity field:

$$v_g(x_t, t, c) = wv(x_t, t \mid c) + (1 - w)v(x_t, t), \tag{23}$$

where $v(x_t, t) = \mathbb{E}_{p(x_0, x_1 | x_t)}[\alpha'_t x_0 + \beta'_t x_1]$ denotes the unconditional marginal velocity field, $c$ represents conditions (e.g., class labels), $v(x_t, t \mid c) = \mathbb{E}_{p(x_0, x_1 | x_t, c)}[\alpha'_t x_0 + \beta'_t x_1]$ denotes the conditional marginal velocity field, and $w$ is the CFG strength. To ensure our conditional model serves as the solution function for the velocity ODE defined by this guided field (which depends purely on the data distribution and is model-independent), it should satisfy the following equations:

$$f_\theta(x_t, t, t, c) = x_t, \tag{24}$$

$$\partial_1 f_\theta(x_t, t, s, c)v_g(x_t, t, c) + \partial_2 f_\theta(x_t, t, s, c) = 0. \tag{25}$$

Analogous to the unconditional setting, we define the guided Flow Matching loss $\mathcal{L}_{\text{FM}}^g(\theta)$ as

$$\mathcal{L}_{\text{FM}}^g(\theta) = \mathbb{E}_{t, x_t, c}\left[\frac{w_{\text{FM}}(t, \text{MSE})}{n} \left\| \partial_2 a(t,t)x_t + \partial_2 b(t,t)F_\theta(x_t, t, t, c) - v_g(x_t, t, c) \right\|_2^2\right], \tag{26}$$

and the guided solution consistency loss $\mathcal{L}_{\text{SCM}}^g(\theta)$ as

$$\mathcal{L}_{\text{SCM}}^g(\theta) = \mathbb{E}_{t, l, s, x_t, c}\left[\frac{w_{\text{SCM}}(t, l, s, \text{MSE})}{n} \left\| f_\theta(x_t, t, s, c) - f_{\theta^-}\left(x_t + v_g(x_t, t, c)(l - t), l, s, c\right) \right\|_2^2\right], \tag{27}$$

where $n$ is the data dimension, $\theta^-$ denotes the stop-gradient operator applied to the targets, and the adaptive weighting functions remain consistent with the unconditional case. The total guided training loss $\mathcal{L}_g(\theta)$ is defined as their linear combination $\lambda \mathcal{L}_{\text{FM}}^g(\theta) + (1 - \lambda)\mathcal{L}_{\text{SCM}}^g(\theta)$.

However, since the unconditional velocity field is generally intractable during conditional training, $v_g(x_t, t, c)$ is not directly accessible. To address this, we train the network to concurrently predict

the unconditional velocity field. Specifically, we randomly replace the condition $c$ with an empty label $\phi$ (with a probability of $0.1$) and update the model using the unconditional Flow Matching loss (Eq. 17) and the unconditional solution consistency loss (Eq. 20). Consequently, we can approximate $v(x_t, t)$ using the model prediction $v_{\text{uncond}} = \partial_2 a(t,t)x_t + \partial_2 b(t,t)F_{\theta^-}(x_t, t, t, \phi)$.

For data points with a non-empty condition $c$, we compute the guided losses by substituting the velocity term $v_g(x_t, t, c)$ in Eq. 26 and Eq. 27 with the estimator $w(\alpha'_t x_0 + \beta'_t x_1) + (1-w)v_{\text{uncond}}$, where $w$ is the CFG strength. Here, $\alpha'_t x_0 + \beta'_t x_1$ is used to replace the intractable conditional marginal velocity field $v(x_t, t \mid c)$, similar to the unconditional formulation.

Notably, the term $w(\alpha'_t x_0 + \beta'_t x_1) + (1-w)v_{\text{uncond}}$ typically exhibits higher variance than the original term $\alpha'_t x_0 + \beta'_t x_1$, primarily due to the scaling effect of $w$. This increased variance can hinder training convergence and degrade final performance. We observe that the model learns the guided velocity field via Eq. 27 when conditioned on $c$. Therefore, we can obtain a model-predicted guided velocity $v_{\text{guided}}$ via $\partial_2 a(t,t)x_t + \partial_2 b(t,t)F_{\theta^-}(x_t, t, t, c)$, and then employ

$$v_{\text{mix}} = m(w(\alpha'_t x_0 + \beta'_t x_1) + (1-w)v_{\text{uncond}}) + (1-m)v_{\text{guided}} \tag{28}$$

to approximate the target $v_g(x_t, t, c)$ (with expectations also taken over $(x_0, x_1)$), where $0 < m \leq 1$ acts as a velocity mixing ratio. Since the stochastic term $\alpha'_t x_0 + \beta'_t x_1$ is the dominant source of variance, using a small $m$ effectively mitigates this issue, as the model-predicted $v_{\text{guided}}$ possesses significantly lower variance.

The inference process of our model is straightforward. Since $f_\theta(x_t, t, s) = a(t,s)x_t + b(t,s)F_\theta(x_t, t, s)$ is a neural solution function to the velocity ODE, we only need to sample $x_1 \sim \mathcal{N}(0, I_n)$ and apply this function with $t = 1$ and $s = 0$ to obtain the clean data in a single step. Furthermore, similar to consistency models (Song et al., 2023), our model supports multi-step sampling by adding noise to the predicted data and recursively applying the solution function.

## 5 EXPERIMENTS

### 5.1 SETTINGS

We conduct our major experiments on the ImageNet $256\times256$ dataset (Deng et al., 2009). SoFlow models operate within the latent space of a pre-trained VAE [1](Rombach et al., 2022), which is a common practice in recent works (Peebles & Xie, 2023; Frans et al., 2024; Geng et al., 2025). The tokenizer converts $256\times256$ images into a $32\times32\times4$ latent representation. We assess generation quality using the Fréchet Inception Distance (FID) (Heusel et al., 2017), computed over a set of 50,000 generated samples. To evaluate computational efficiency, we report the number of function evaluations (NFE), with a particular focus on the single-step (1-NFE) scenario. All models presented are trained from scratch. In addition to the standard time variable $t$, our model incorporates an additional time variable $s$. We provide this to the networks by feeding the positional embeddings (Vaswani et al., 2017) of their difference, $s - t$. For more implementation details, please refer to Appendix A.

### 5.2 ABLATION STUDY

We conduct various ablation studies to determine the optimal hyperparameters for SoFlow models. Following the methodology of Geng et al. (2025), we utilize the DiT-B/4 architecture in our ablation experiments, which features a "base" sized Diffusion Transformer with a patch size of 4. We train our models for 400K iterations. For performance reference, the original DiT-B/4 (Peebles & Xie, 2023) achieves an FID-50K of 68.4 with 250-NFE sampling. MeanFlow (Geng et al., 2025) reports an FID-50K of 61.06 with 1-NFE sampling, and the authors claim to have reproduced SiT-B/4 (Ma et al., 2024) with an FID-50K of 58.9 using 250-NFE sampling. The FID-50K scores mentioned here are without CFG. Our ablation studies are conducted in two groups: a first group of three experiments without CFG and a second group of three with CFG. Table 1 presents our results, which are analyzed as follows:

---

[1]SD-VAE: `https://huggingface.co/stabilityai/sd-vae-ft-mse`

| $r(k, K)$ | FID-50K |
|---|---|
| Exponential | **58.57** |
| Cosine | 60.17 |
| Linear | 59.01 |
| Constant | 59.52 |

(a) $l \to t$ schedule $r(k, K)$

| $\lambda$ | FID-50K |
|---|---|
| 0% | 66.36 |
| 25% | 61.42 |
| 50% | 59.61 |
| 75% | **58.57** |

(b) Flow Matching data ratio $\lambda$

| $p$ | FID-50K |
|---|---|
| 0.0 | 68.25 |
| 0.5 | 59.04 |
| 1.0 | **58.57** |
| 1.5 | 63.72 |

(c) Loss weighting coefficient $p$

| $\alpha_t, \beta_t$ and $a(t, s), b(t, s)$ | FID-50K |
|---|---|
| Linear, Euler | **11.59** |
| Linear, Trigonometric | 17.50 |
| Trigonometric, Euler | 13.12 |
| Trigonometric, Trigonometric | 19.01 |

(d) Noising schedule and parameterization

| $w$ | FID-50K |
|---|---|
| 1.5 | 34.47 |
| 2.0 | 20.35 |
| 2.5 | 14.60 |
| 3.0 | **11.59** |

(e) CFG strength $w$

| $m$ | FID-50K |
|---|---|
| 0.25 | **11.59** |
| 0.5 | 13.94 |
| 0.75 | 16.29 |
| 1.0 | 17.22 |

(f) Velocity mix ratio $m$

Table 1: **Ablation studies on 1-NFE ImageNet 256×256 class-conditional generation.** All models have 131M parameters and are trained with a batch size of 256 for 400K iterations from scratch.

$l \to t$ **Schedule** $r(k, K)$. We compare exponential, cosine, linear, and constant schedules in our experiments (Table 1a). The FID-50K results show that the speed of the $l \to t$ transition during the training process has a relatively small influence on performance. We choose the best exponential schedule following previous works (Song & Dhariwal, 2023; Geng et al., 2024).

**Flow Matching Data Ratio** $\lambda$. Although our solution consistency loss alone is sufficient to enable one-step generation (corresponding to a 0% ratio in Table 1b), experimental results show that incorporating a large ratio of Flow Matching loss is effective for improving performance.

**Loss Weighting Coefficient** $p$. We observe that the choice of loss metric has a significant influence on the performance of one-step generation, which aligns with previous works (Song & Dhariwal, 2023; Geng et al., 2024; 2025; Dao et al., 2025). As shown in Table 1c, the original mean squared loss with $p = 0$ performs much worse than $p = 0.5$ or $p = 1$ settings.

**Noising Schedule and Parameterization.** Our method is compatible with different Flow Matching noising schedules and solution function parameterizations. As presented in Table 1d, experiments show that the linear noising schedule $x_t = (1 - t)x_0 + tx_1$ along with the Euler parameterization $f_\theta(x_t, t, s) = x_t + (s - t)F_\theta(x_t, t, s)$ performs best, compared to the trigonometric noising schedule $x_t = \cos(\frac{\pi}{2}t)x_0 + \sin(\frac{\pi}{2}t)x_1$ and its corresponding parameterization $f_\theta(x_t, t, s) = \cos\left(\frac{\pi}{2}(s - t)\right) x_t + \sin\left(\frac{\pi}{2}(s - t)\right) F_\theta(x_t, t, s)$.

**CFG Strength** $w$. Unlike diffusion models, our model enables CFG during the training stage, so the resulting 1-NFE inference is guided without additional inference-time CFG. Experiments in Table 1e demonstrate that CFG can also greatly improve the generation quality for our model.

**Velocity Mix Ratio** $m$. Experiments (Table 1f) show that a relatively small $m$ is beneficial for performance. This can be explained by the fact that the CFG guidance strength $w$ amplifies the variance in the velocity term, and a smaller $m$ can suppress this variance by partially replacing the velocity term with the model's prediction of the guided velocity field.

## 5.3 COMPARISON WITH OTHER WORKS

**ImageNet 256×256 Results.** We begin by analyzing the 1-NFE FID-50K results of our model across various model sizes (numbers of parameters), as shown in Figure 3. Our model's 1-NFE performance gradually improves with an increase in model parameters, which is consistent with previous observations on Diffusion Transformers (Peebles & Xie, 2023; Ma et al., 2024; Geng et al., 2025).

Next, we compare our model's one-step generation performance against previous models, with the results summarized in Table 2. To make relatively fair comparisons, we train our model by setting the batch size to 256 and the training epochs to 240, consistent with the MeanFlow models (Geng et al.,

| method | epochs | params | NFE | FID↓ |
|---|---|---|---|---|
| ***Generative Adversarial Networks*** | | | | |
| BigGAN (Brock et al., 2018) | - | 112M | 1 | 6.95 |
| StyleGAN-XL (Sauer et al., 2022) | - | 166M | 1 | **2.30** |
| GigaGAN (Kang et al., 2023) | - | 569M | 1 | 3.45 |
| ***Masked and Autoregressive Models*** | | | | |
| Mask-GIT (Chang et al., 2022) | 555 | 227M | 8 | 6.18 |
| MagViT-v2 (Yu et al., 2023) | 1080 | 307M | 64 | 1.78 |
| LlamaGen-XL (Sun et al., 2024) | 300 | 775M | 576 | 2.62 |
| VAR (Tian et al., 2024) | 350 | 2.0B | 10 | 1.80 |
| MAR (Li et al., 2024) | 800 | 943M | 64 | **1.55** |
| RandAR-XL (Pang et al., 2025) | 300 | 775M | 256 | 2.22 |
| ***Multi-step Diffusion Models*** | | | | |
| LDM-4-G (Rombach et al., 2022) | 170 | 395M | 250×2 | 3.60 |
| MDTv2 (Gao et al., 2023) | 700 | 676M | 250×2 | 1.63 |
| DiT-XL/2 (Peebles & Xie, 2023) | 1400 | 675M | 250×2 | 2.27 |
| SiT-XL/2 (Ma et al., 2024) | 1400 | 675M | 250×2 | 2.06 |
| FlowDCN-XL/2 (Wang et al., 2024) | 400 | 675M | 250×2 | 2.00 |
| SiT-REPA-XL/2 (Yu et al., 2024) | 800 | 675M | 250×2 | **1.42** |
| ***Few-step Diffusion Models*** | | | | |
| iCT-XL/2[†] (Song & Dhariwal, 2023) | - | 675M | 1 / 2 | 34.24 / 20.30 |
| Shortcut-XL/2 (Frans et al., 2024) | 250 | 675M | 1 / 4 | 10.60 / 7.80 |
| IMM-XL/2 (Zhou et al., 2025) | 3840 | 675M | 1×2 / 2×2 | 7.77 / 3.99 |
| MeanFlow-B/2 (Geng et al., 2025) | 240 | 131M | 1 | 6.17 |
| MeanFlow-M/2 (Geng et al., 2025) | 240 | 308M | 1 | 5.01 |
| MeanFlow-L/2 (Geng et al., 2025) | 240 | 459M | 1 | 3.84 |
| MeanFlow-XL/2 (Geng et al., 2025) | 240 | 676M | 1 / 2 | 3.43 / 2.93 |
| SoFlow-B/2 | 240 | 131M | 1 / 2 | 4.85 / 4.24 |
| SoFlow-M/2 | 240 | 308M | 1 / 2 | 3.73 / 3.42 |
| SoFlow-L/2 | 240 | 459M | 1 / 2 | 3.20 / 2.90 |
| SoFlow-XL/2 | 240 | 676M | 1 / 2 | **2.96 / 2.66** |

Table 2: **FID-50K results for class-conditional generation on ImageNet 256×256.** ×2 denotes an NFE of 2 per sampling step incurred by CFG. Entries in the format "1 / 2" indicate that the corresponding FID scores are reported for 1-NFE and 2-NFE sampling, respectively. [†] Results as reported in Zhou et al. (2025).

2025). Experimental results show that our model consistently outperforms MeanFlow models across all evaluated model sizes when trained from scratch. Specifically, for smaller models with DiT-B/2 and DiT-M/2 architectures, our model demonstrates significant improvements, achieving FID-50K scores of 4.85 and 3.73, respectively. Our model also achieves superior FID-50K values for larger architectures, namely DiT-L/2 and DiT-XL/2, reaching 3.20 and 2.96, respectively. Notably, our models employ CFG during training, which enables generation with exactly 1-NFE during inference, similar to MeanFlow models. Furthermore, our XL/2 model achieves a strong 2-NFE FID-50K score of 2.66, surpassing the 2.93 achieved by MeanFlow-XL/2.

In terms of computational efficiency, since our method obviates the need for JVP computation, it benefits from lower GPU memory usage and faster training speeds compared to MeanFlow models. Please refer to Appendix D for a detailed comparison.

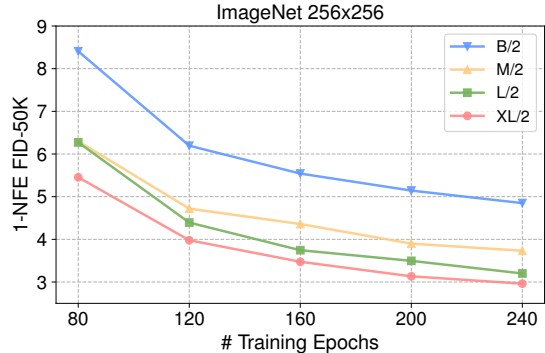

Figure 3: **FID-50K performance of our models.** The plot shows the FID-50K performance of our models with varying numbers of parameters, all trained from scratch on the ImageNet 256×256 dataset. We apply CFG during training and report the FID-50K scores of generated images using a 1-NFE sampling process. The results consistently demonstrate that the performance of our models improves as the model size increases.

**CIFAR-10 Results.** In Table 3, we report unconditional generation results on the CIFAR-10 (Krizhevsky et al., 2009) dataset, where performance is measured by the FID-50K metric with 1-NFE sampling. For our model, we adopt the U-Net architecture (Ronneberger et al., 2015) developed from Song et al. (2020), aligning with prior works. Our method is applied directly to the pixel space, with a resolution of 32×32. For more implementation details, please refer to Appendix A. Our method achieves competitive performance compared to prior approaches on this dataset.

| method | NFE | FID |
|---|---|---|
| iCT (Song & Dhariwal, 2023) | 1 | **2.83** |
| ECT (Geng et al., 2024) | 1 | 3.60 |
| sCT (Lu & Song, 2024) | 1 | 2.97 |
| IMM (Zhou et al., 2025) | 1 | 3.20 |
| MeanFlow (Geng et al., 2025) | 1 | 2.92 |
| SoFlow | 1 | 2.86 |

Table 3: **FID-50K results for unconditional generation on CIFAR-10.**

## 6 CONCLUSION

We have presented SoFlow, a simple yet effective framework for one-step generative modeling. Our approach directly learns the solution function of the velocity ODE, enabling single-step sampling without iterative solvers. By leveraging a bi-time formulation and a hybrid training objective combining a Flow Matching loss and a solution consistency loss, SoFlow naturally supports CFG during training and avoids JVP calculations that are not well-optimized in deep learning frameworks like PyTorch. Our method demonstrates competitive performance on the class-conditional ImageNet 256×256 generation task, outperforming MeanFlow models when trained from scratch under the same settings.

ACKNOWLEDGMENTS

We thank Kaiming He for helpful discussions. We gratefully acknowledge the use of the Neuronic GPU cluster maintained by the Department of Computer Science at Princeton University. This work was substantially performed using Princeton Research Computing resources, a consortium led by the Princeton Institute for Computational Science and Engineering (PICSciE) and Research Computing at Princeton University. We also thank Princeton Language and Intelligence (PLI) for H100 GPU support for this work.

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

# APPENDIX

## A    IMPLEMENTATION DETAILS

We first provide a detailed description of the different scheduling strategies used to determine the intermediate time variable $l$ during training. Recall that $l$ is computed from $t$ and $s$ as follows:

$$l = t + (s - t) \times r(k, K),$$

where $k, K$ represent the current and total training steps, and the function $r(k, K)$ is a monotonically decreasing function that controls the progression of $l$ from an initial value near $t$ towards $t$ itself as training advances. Below we specify the four schedules compared in our ablation studies:

**Exponential Schedule:**

$$r(k, K) = r_{\text{init}} \times \left( \frac{r_{\text{end}}}{r_{\text{init}}} \right)^{\frac{k}{K}},$$

**Cosine Schedule:**

$$r(k, K) = r_{\text{end}} + (r_{\text{init}} - r_{\text{end}}) \times \frac{1}{2} \left( 1 + \cos \left( \pi \cdot \frac{k}{K} \right) \right),$$

**Linear Schedule:**

$$r(k, K) = r_{\text{init}} + (r_{\text{end}} - r_{\text{init}}) \times \frac{k}{K},$$

**Constant Schedule:**

$$r(k, K) = r_{\text{end}}.$$

In all cases, $l$ is clamped to satisfy $l < t - 10^{-4}$ to ensure numerical stability; similarly, we also clamp $s$ to satisfy $s < t - 10^{-4}$. The initial value $r_{\text{init}}$ and end value $r_{\text{end}}$ are hyperparameters controlling the starting and final relative position between $l$ and $t$. We now detail the training and architectural configurations for our models on two benchmark datasets. All experiments are run on NVIDIA H100 GPUs.

**ImageNet 256×256.**    We employ the AdamW optimizer (Loshchilov & Hutter, 2017) with a constant learning rate of $1 \times 10^{-4}$ and betas set to $(0.9, 0.99)$, without learning rate decay or weight decay. Following standard practice, we evaluate model performance using Exponential Moving Average (EMA) with a decay rate of 0.9999. For time sampling, the logit-normal distribution parameters are set as: $\mu_{\text{FM}} = -0.2, \sigma_{\text{FM}} = 1.0$; $\mu_t = 0.2, \sigma_t = 0.8$; $\mu_s = -1.0, \sigma_s = 0.8$. The Flow Matching data ratio is $75\%$ and the adaptive loss weight coefficient $p = 1.0$. The schedule parameters are $r_{\text{init}} = \frac{1}{10}$ and $r_{\text{end}} = \frac{1}{500}$. We use a linear noising schedule with Euler parameterization. The CFG strength $w$ is set to 2.5, 2.25, 2.0, and 2.0 for the B/2, M/2, L/2, and XL/2 models, respectively, while the velocity mix ratio $m$ is set to 0.25 for all models. Following common practice, we gradually decay the CFG strength to 1.0 in high-noise regions, setting the decay threshold at $t > 0.8$ for ablation studies and $t > 0.75$ for the main experiments. Specifically, the decay speed is determined by the function $1 - \exp \left( -\frac{1}{40} \frac{t'}{1-t'} \right)$, where $t' = \min \left( \frac{1-t}{1-t_{\text{decay}}}, 1 - 10^{-6} \right)$ for $t > t_{\text{decay}}$. Here, $t_{\text{decay}}$ denotes the threshold time and $10^{-6}$ is used to ensure numerical stability. We employ this specific smoothing function rather than a direct decay to ensure that the guided velocity field remains continuously differentiable. Finally, architectural details are provided in Table 4.

**CIFAR-10.** Training uses a batch size of 1024 for 800K iterations, consistent with MeanFlow. We adopt the RAdam (Liu et al., 2019) optimizer with a learning rate of $1 \times 10^{-4}$, following Song & Dhariwal (2023); Geng et al. (2024). We evaluate model performance using EMA with a decay rate of 0.9999. Time sampling parameters are: $\mu_{\text{FM}} = -0.9, \sigma_{\text{FM}} = 1.6$; $\mu_t = -0.9, \sigma_t = 1.6$; $\mu_s = -4.0, \sigma_s = 1.6$. The Flow Matching data ratio is $0\%$ with $p = 0.75$. Schedule parameters $r_{\text{init}}$ and $r_{\text{end}}$ are set to 1.0 and $\frac{1}{3200}$ respectively. Linear noising schedule and Euler parameterization are used. Our data augmentation setup follows Karras et al. (2022), where vertical flipping and rotation augmentations are disabled.

| Architectures | SoFlow-B/4 | SoFlow-B/2 | SoFlow-M/2 | SoFlow-L/2 | SoFlow-XL/2 |
|---|---|---|---|---|---|
| Parameters (M) | 131 | 131 | 308 | 459 | 676 |
| FLOPs ($\approx$ G) | 5.6 | 23.0 | 54.0 | 81.0 | 119.0 |
| Depth | 12 | 12 | 16 | 24 | 28 |
| Hidden dimension | 768 | 768 | 1024 | 1024 | 1152 |
| Attention heads | 12 | 12 | 16 | 16 | 16 |
| Patch size | 4×4 | 2×2 | 2×2 | 2×2 | 2×2 |
| Training epochs | 80 | 240 | 240 | 240 | 240 |

Table 4: **Configurations on the ImageNet 256×256 dataset.** We detail the specifications for our models, which are based on the Diffusion Transformer (Peebles & Xie, 2023) architecture. The configurations scale from a 131M parameter model (B/4) to a 676M parameter model (XL/2) to evaluate performance across different capacities.

## B  MORE VISUAL SAMPLES

We present additional visual results generated by our models for the ImageNet 256×256 and CIFAR-10 datasets in Figure 4 and Figure 5, respectively.

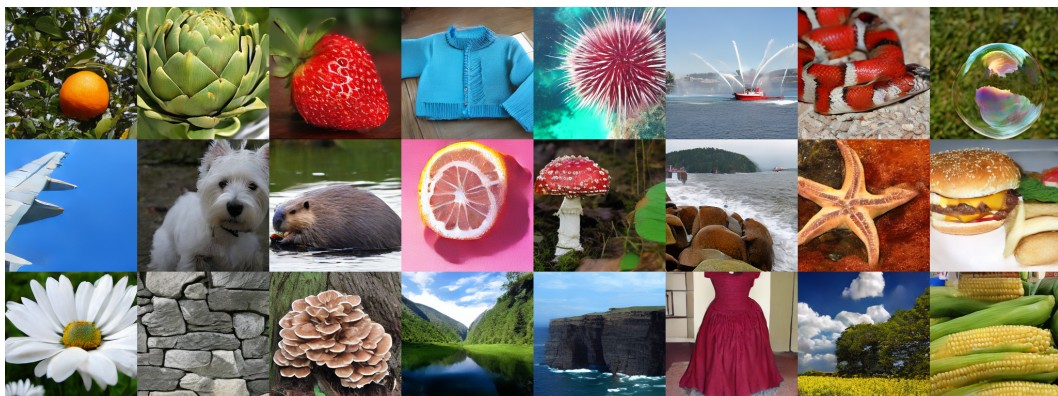

Figure 4: Curated conditional 1-NFE samples by our XL/2 model on the ImageNet 256×256 dataset.

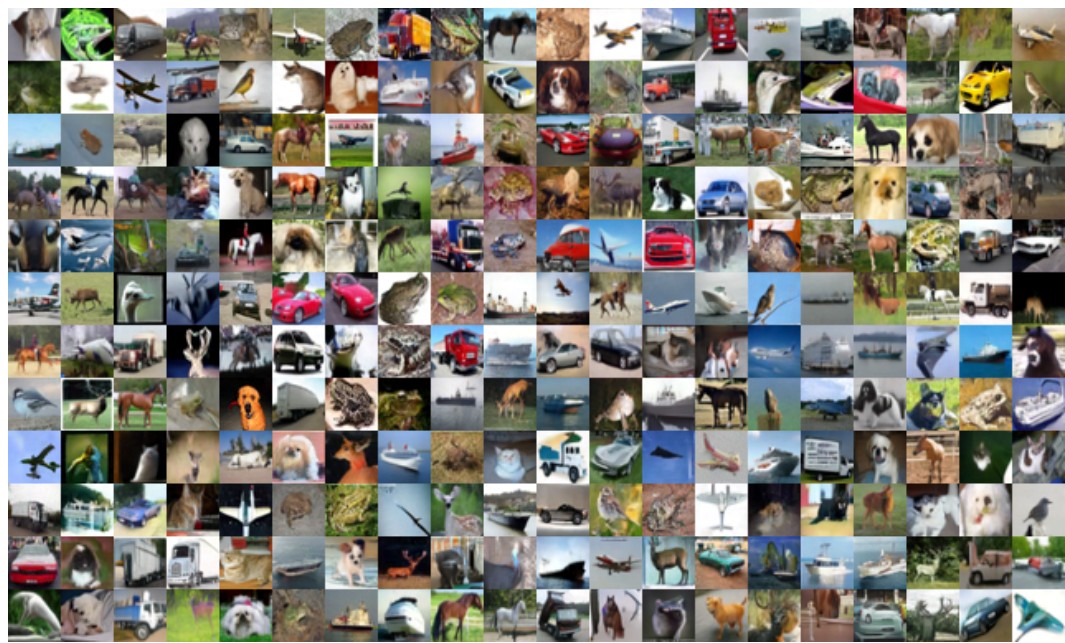

Figure 5: Uncurated 1-NFE unconditional samples by our U-Net model on the CIFAR-10 dataset.

## C    PSEUDO-CODE FOR GUIDED TRAINING PROCESS

To facilitate a deeper understanding of the implementation, we detail the pseudo-code for the guided training process in Algorithm 1.

---
**Algorithm 1** Train Solution Flow Models with CFG
---

**Input:** model $f_\theta(x_t, t, s, c)$, Flow Matching data ratio $\lambda$, CFG strength $w$, velocity mix ratio $m$

Sample a data batch: $x_0, c \sim p_{data}(x_0, c)$

Split the data batch for two losses by $\lambda$: $(x_0^{\text{FM}}, x_0^{\text{SCM}}) = x_0, (c^{\text{FM}}, c^{\text{SCM}}) = c$

Sample $t_{\text{FM}}, t_{\text{SCM}}, l_{\text{SCM}}, s_{\text{SCM}}$ according to subsection 4.2

Compute $x_t^{\text{FM}}, v_t^{\text{FM}}, x_t^{\text{SCM}}, v_t^{\text{SCM}}$ using standard Flow Matching framework

Compute $v_{\text{mix}}^{\text{FM}}$ and $v_{\text{mix}}^{\text{SCM}}$ with Eq. 28 with $w$ and $m$

Randomly replace $v_{\text{mix}}^{\text{FM}}$ and $v_{\text{mix}}^{\text{SCM}}$ with $v_t^{\text{FM}}$ and $v_t^{\text{SCM}}$ with a CFG drop rate 0.1

Replace $c^{\text{FM}}$ and $c^{\text{SCM}}$ with the empty label $\phi$ correspondingly

Compute $\mathcal{L}_{\text{FM}}^g(\theta)$ according to Eq. 26 by replacing the $v_g(x_t, t, c)$ term with $v_{\text{mix}}^{\text{FM}}$

Compute $\mathcal{L}_{\text{SCM}}^g(\theta)$ according to Eq. 27 by replacing the $v_g(x_t, t, c)$ term with $v_{\text{mix}}^{\text{SCM}}$

Update $f_\theta$ via gradient descent according to $\mathcal{L}_g(\theta) = \lambda \mathcal{L}_{\text{FM}}^g(\theta) + (1 - \lambda)\mathcal{L}_{\text{SCM}}^g(\theta)$

---

## D    TRAINING EFFICIENCY

It is important to note that GPU memory consumption and training speed rely heavily on implementation details, hardware, and frameworks. To ensure a fair comparison, we evaluate the training costs of MeanFlow and SoFlow models using an identical PyTorch codebase, differing only by replacing our solution consistency loss with the MeanFlow loss.

The results are presented in Table 5. Currently, PyTorch's memory-efficient attention implementation supports standard forward and backward passes but lacks support for the Jacobian-vector product (JVP) computation required by MeanFlow. Consequently, MeanFlow must rely on the standard "math" attention backend. In contrast, our solution consistency loss is fully compatible with

memory-efficient attention. As a result, our model reduces peak GPU memory usage by approximately 31% and increases training speed by 23% compared to MeanFlow, demonstrating significant gains in computational efficiency.

|  | MeanFlow (Math Attn.) | SoFlow (Math Attn.) | SoFlow (Efficient Attn.) |
|---|---|---|---|
| GPU Memory | 51.45 GB | 38.95 GB | 35.44 GB |
| Training Speed | 2.39 iters/sec | 2.84 iters/sec | 2.94 iters/sec |

Table 5: Training Speed and GPU peak memory usage comparison.

## E  THEORETICAL ANALYSIS

In this section, we provide additional theoretical analysis to demonstrate that our model can effectively learn the global ground truth solution function based on our local consistency objectives.

Specifically, we analyze the error between our model $f_\theta(x_t, t, s)$ and the ground truth $f(x_t, t, s)$. Using the fundamental theorem of calculus, the error can be expressed as:

$$f(x_t, t, s) - f_\theta(x_t, t, s) = \int_t^s \frac{\partial}{\partial l}(f_\theta(f(x_t, t, l), l, s))dl. \tag{29}$$

This equality holds because the boundary terms satisfy $f_\theta(f(x_t, t, t), t, s) = f_\theta(x_t, t, s)$ and $f_\theta(f(x_t, t, s), s, s) = f(x_t, t, s)$. Applying the chain rule, the integrand term can be expanded as:

$$\frac{\partial}{\partial l}(f_\theta(f(x_t, t, l), l, s)) = \partial_1 f_\theta(f(x_t, t, l), l, s)\partial_3 f(x_t, t, l) + \partial_2 f_\theta(f(x_t, t, l), l, s). \tag{30}$$

Since $f(x_t, t, s)$ is the ground truth solution function of the velocity ODE defined by $v(x_t, t)$, we have $\partial_3 f(x_t, t, l) = v(f(x_t, t, l), l)$. We define the residual term $R_\theta(x_t, t, s)$ as:

$$R_\theta(x_t, t, s) = \partial_1 f_\theta(x_t, t, s)v(x_t, t) + \partial_2 f_\theta(x_t, t, s), \tag{31}$$

where we refer to $\|R_\theta(x_t, t, s)\|_2$ as the partial differential equation (PDE) error. Substituting this into Eq. 30, we simplify the expression to:

$$\frac{\partial}{\partial l}(f_\theta(f(x_t, t, l), l, s)) = R_\theta(f(x_t, t, l), l, s). \tag{32}$$

Combining this with Eq. 29, we obtain:

$$f(x_t, t, s) - f_\theta(x_t, t, s) = \int_t^s R_\theta(f(x_t, t, l), l, s)dl. \tag{33}$$

During the training process, our model minimizes the difference loss $\|f_\theta(x_t, t, s) - f_{\theta^-}(x_t + v(x_t, t)(l - t), l, s)\|_2$ with an adaptive scaling function. To derive an upper bound for the error between our model's prediction and the ground truth, we assume that after training, the loss term $\|f_\theta(x_t, t, s) - f_{\theta^-}(x_t + v(x_t, t)(l - t), l, s)\|_2$ is uniformly bounded by $e_{max}|t - l|$, where $e_{max}$ is a constant and $|t - l|$ represents the magnitude of the error.

Assuming $f_\theta(x_t, t, s)$ and $v(x_t, t)$ are twice-continuously differentiable with bounded second-order derivatives, we can apply the Taylor expansion to bound the discrepancy:

$$\|f_\theta(x_t, t, s) - f_{\theta^-}(x_t + v(x_t, t)(l - t), l, s) - R_\theta(x_t, t, s)(t - l)\|_2 \le H(t - l)^2. \tag{34}$$

This implies the remainder is uniformly bounded by $H(t - l)^2$ given the bounded second-order derivatives. Using the triangle inequality, we combine these bounds:

$$\|R_\theta(x_t, t, s)\|_2 \le e_{max} + H|t - l| = e_{max} + H|s - t|r(k, K), \tag{35}$$

where $r(k, K) = \frac{l-t}{s-t}$ is the decreasing schedule function determining $l$ as discussed in our paper, and $k, K$ denote the current and total training steps, respectively. For simplicity, we denote this

upper bound as $\delta$, i.e., $\|R_\theta(x_t, t, s)\|_2 \leq \delta$. Finally, substituting this back into Eq. 33, we arrive at the global error bound:

$$\|f(x_t, t, s) - f_\theta(x_t, t, s)\|_2 \leq |\int_t^s \|R_\theta(f(x_t, t, l), l, s)\|_2 dl| \leq |s - t|\delta. \tag{36}$$

As the training loss upper bound $e_{\max}$ and the schedule function value $r(k, K)$ decrease sufficiently during training, our model $f_\theta(x_t, t, s)$ effectively approximates the solution function $f(x_t, t, s)$ with low error.

Furthermore, we provide a theoretical analysis to show that our objective not only enables our model to learn the ground truth solution function but also implicitly minimizes the ODE error $\|\partial_3 f_\theta(x_t, t, s) - v(f_\theta(x_t, t, s), s)\|_2$. To derive a bound for the ODE error, we make three mild assumptions. First, we assume that after training, the residual satisfies $\|R_\theta(x_t, t, s)\|_2 \leq \delta$ for all $x \in \mathbb{R}^n$ and $0 \leq s \leq t \leq 1$, where $\delta = e_{\max} + H|s - t|r(k, K)$ as discussed above. Second, we assume that $v(x_t, t)$ is continuously differentiable and Lipschitz continuous with respect to $x_t$, i.e., $\|v(x_t, t) - v(y_t, t)\|_2 \leq L_v\|x_t - y_t\|_2$. Third, we assume that $R_\theta(x_t, t, s)$ is continuously differentiable and its partial derivative with respect to $s$ is Lipschitz continuous; that is, $\|\partial_3 R_\theta(x_t, t, s_1) - \partial_3 R_\theta(x_t, t, s_2)\|_2 \leq L|s_1 - s_2|$, where $L$ is the Lipschitz constant. We first prove the following lemma:

**Lemma.** Under the stated assumptions, $\partial_3 R_\theta(x_t, t, s)$ is uniformly bounded by:

$$\|\partial_3 R_\theta(x_t, t, s)\|_2 \leq 2\sqrt{L\delta}. \tag{37}$$

**Proof.** For brevity, we denote $R_\theta(x_t, t, s)$ as $g(s) \in \mathbb{R}^n$. We aim to bound $\|g'(s)\|_2$ given that $\|g(s)\|_2 \leq \delta$ and $\|g'(s_1) - g'(s_2)\|_2 \leq L|s_1 - s_2|$. According to the fundamental theorem of calculus, we have:

$$g(s + h) - g(s) = \int_0^1 g'(s + \tau h)h \, d\tau. \tag{38}$$

We can rewrite the term $g'(s)h$ as follows:

$$
\begin{aligned}
g'(s)h &= g'(s)h + g(s + h) - g(s) - \int_0^1 g'(s + \tau h)h \, d\tau \\
&= g(s + h) - g(s) - \int_0^1 (g'(s + \tau h) - g'(s))h \, d\tau.
\end{aligned}
\tag{39}
$$

Taking the Euclidean norm on both sides and applying the triangle inequality yields:

$$|h|\|g'(s)\|_2 \leq \|g(s + h)\|_2 + \|g(s)\|_2 + \left\|\int_0^1 (g'(s + \tau h) - g'(s))h \, d\tau\right\|_2. \tag{40}$$

Using the assumption $\|g(\cdot)\|_2 \leq \delta$ and the Lipschitz continuity of $g'(s)$, we can bound the terms using the triangle inequality for integrals:

$$|h|\|g'(s)\|_2 \leq \delta + \delta + \int_0^1 \|g'(s + \tau h) - g'(s)\|_2|h| \, d\tau \tag{41}$$

$$\leq 2\delta + \int_0^1 L|\tau h||h| \, d\tau \tag{42}$$

$$= 2\delta + Lh^2 \int_0^1 \tau \, d\tau \tag{43}$$

$$= 2\delta + \frac{Lh^2}{2}. \tag{44}$$

Note that this inequality holds for any $h \neq 0$. We consider the case where $h > 0$ to form an upper bound. Dividing by $|h|$, we obtain:

$$\|g'(s)\|_2 \leq \frac{2\delta}{h} + \frac{Lh}{2}, \quad \forall h > 0. \tag{45}$$

To find the tightest bound, we minimize the right-hand side with respect to $h$. The minimum occurs when the derivative with respect to $h$ is zero (since the second derivative is positive), yielding:

$$-\frac{2\delta}{h^2} + \frac{L}{2} = 0 \implies h^2 = \frac{4\delta}{L} \implies h = 2\sqrt{\frac{\delta}{L}}. \tag{46}$$

Substituting this optimal $h$ back into the inequality gives:

$$\|g'(s)\|_2 \leq \frac{2\delta}{2\sqrt{\delta/L}} + \frac{L}{2}\left(2\sqrt{\frac{\delta}{L}}\right) = \sqrt{L\delta} + \sqrt{L\delta} = 2\sqrt{L\delta}. \tag{47}$$

This concludes the proof of the lemma.

Next, taking the partial derivatives with respect to $s$ in Eq. 33, we have:

$$\partial_3 f(x_t, t, s) - \partial_3 f_\theta(x_t, t, s) = R_\theta(f(x_t, t, s), s, s) + \int_t^s \partial_3 R_\theta(f(x_t, t, l), l, s)dl. \tag{48}$$

Using this equation, we can write the ODE residual vector as follows:

$$\partial_3 f_\theta(x_t, t, s) - v(f_\theta(x_t, t, s), s)$$
$$= \partial_3 f(x_t, t, s) - v(f_\theta(x_t, t, s), s) - R_\theta(f(x_t, t, s), s, s) - \int_t^s \partial_3 R_\theta(f(x_t, t, l), l, s)dl$$
$$= v(f(x_t, t, s), s) - v(f_\theta(x_t, t, s), s) - R_\theta(f(x_t, t, s), s, s) - \int_t^s \partial_3 R_\theta(f(x_t, t, l), l, s)dl. \tag{49}$$

We first use the Lipschitz continuity of $v(x_t, t)$, along with Eq. 36:

$$\|v(f(x_t, t, s), s) - v(f_\theta(x_t, t, s), s)\|_2 \leq L_v\|f(x_t, t, s) - f_\theta(x_t, t, s)\|_2 \leq L_v|s - t|\delta. \tag{50}$$

Since we have a uniform bound $\|R_\theta(x_t, t, s)\|_2 \leq \delta$, it follows that:

$$\|R_\theta(f(x_t, t, s), s, s)\|_2 \leq \delta. \tag{51}$$

Finally, for the integral term, using the triangle inequality and the lemma, we have:

$$\left\|\int_t^s \partial_3 R_\theta(f(x_t, t, l), l, s)dl\right\|_2 \leq \left|\int_t^s \|\partial_3 R_\theta(f(x_t, t, l), l, s)\|_2 dl\right|$$
$$\leq \left|\int_s^t 2\sqrt{L\delta}dl\right| = 2|s - t|\sqrt{L\delta}. \tag{52}$$

Combining these three terms, we obtain:

$$\|\partial_3 f_\theta(x_t, t, s) - v(f_\theta(x_t, t, s), s)\|_2 \leq L_v|s - t|\delta + \delta + 2|s - t|\sqrt{L\delta} = O(\sqrt{\delta}). \tag{53}$$

We have theoretically proved that the ODE error $\|\partial_3 f_\theta(x_t, t, s) - v(f_\theta(x_t, t, s), s)\|_2$ of our model $f_\theta(x_t, t, s)$ is bounded by the PDE error $\|R_\theta(x_t, t, s)\|_2$, where $\delta = e_{\max} + H|s - t|r(k, K)$. During training, the training error $e_{\max}$ is minimized and the schedule function $r(k, K)$ also decreases to a small value. Thus, we conclude that our model implicitly minimizes the ODE error during training with a theoretical guarantee.

## F   LARGE LANGUAGE MODEL USAGE

We only adopt large language models to polish the writing.

