# OpenReview forum: "SoFlow: Solution Flow Models for One-Step Generative Modeling"
_ICLR.cc/2026/Conference — ICLR 2026 Poster_

### Official Review · Reviewer_NydX · 2025-10-17

**Soundness:** 2
**Presentation:** 3
**Contribution:** 2
**Rating:** 4
**Confidence:** 4

**Summary:**

### **Motivation**
- The paper is motivated by the computational inefficiency of recent Consistency Models, which still rely on local supervision using Jacobian–Vector Products (JVPs) to enforce consistency of velocity fields at intermediate timesteps.
- Such methods remain local approximations of ODE dynamics — they match instantaneous velocities rather than learning the global solution function that defines the full trajectory.
- To overcome this, the authors propose directly modeling the ODE solution itself with a neural network, aiming to bypass both the high JVP cost and the reliance on multi-step numerical integration or teacher distillation.


### **Methods**
- Approximates the true ODE solution using a neural network \( f_\theta(x_t, t, s) \) under sufficient conditions ensuring equivalence.
- Introduces **Flow Matching Loss** (for boundary alignment at \( s=t \)) and **Solution Consistency Loss** (for temporal consistency across \( s<t \)).
- Removes the need for JVP computation and integrates Classifier-Free Guidance (CFG) directly into training for guided one-step generation.

### **Contributions**
- Presents a solution-function–based training approach as an alternative to velocity-field supervision in existing flow and consistency models.
- Describes a training-time CFG procedure (including variance-reducing velocity mixing) compatible with one-step inference.
- Reports improved FID-50K on ImageNet 256×256 (3.35 vs. 3.43 for MeanFlow) using the same DiT-based backbone and cifar-10

**Strengths:**

- **[S1] Clear motivation and formulation:** The paper articulates a coherent motivation for moving from locally supervised velocity-field models toward a solution-function approach, addressing computational inefficiency in consistency-based methods.

- **[S2] Consistent empirical improvements:** SoFlow demonstrates FID gains across all MeanFlow model sizes on ImageNet-256×256, while maintaining stable one-step generation, indicating practical viability of the proposed approach

**Weaknesses:**

**[W1] Lack of theoretical grounding in global solution learning**

SoFlow claims to learn the global ODE solution operator (Eq. (6)) directly, but its training relies only on local consistency objectives (eq. (14)). While this condition theoretically implies $ f_\theta = f $, the actual supervision—based on first-order Taylor approximations and stochastic sampling of $(t, l, s)$—remains inherently local, creating a gap between the theoretical formulation and practical optimization.

Traditional Flow Matching constructs a global velocity field via CFM, where local supervision is provably consistent with the true global dynamics. Similarly, Consistency Models enforce locally consistent trajectories using Jacobian-based constraints to form a globally coherent velocity field. In contrast, SoFlow extends this idea to the solution trajectory itself, but without a comparable theoretical foundation.

From a functional approximation perspective, to the best of my knowledge, there is no established theoretical evidence that a neural parametrization of the ODE solution trajectory can recover global behavior solely through such locally defined objectives. As this forms the paper’s main conceptual contribution, providing additional theoretical analysis or empirical validation would help substantiate the claim.

**[W2] Ambiguity in guidance supervision**

Unlike MeanFlow, which learns a local velocity field
$v_\theta(x, t) \approx \mathbb{E}_{p(x_0, x_1|x_t)}[\alpha_t' x_0 + \beta_t' x_1]$ and guarantees global trajectory consistency via explicit numerical integration,

SoFlow directly parameterizes a solution operator
$f_\theta(x_t, t, s) \approx x_t + \int_t^s v(f(x_t, t, u), u)du.$

This global mapping is nonlocal—it depends on the entire integral path of $v$—and therefore the training problem is fundamentally different in nature.

The proposed solution consistency loss (Eq. 14) is based on a first-order Taylor approximation:
$f_\theta(x_t, t, s) - f_\theta(x_t + v(x_t, t)(l - t), l, s) \propto \partial_1 f_\theta(x_t, t, s) v(x_t, t) + \partial_2 f_\theta(x_t, t, s),$
which supervises local temporal consistency between nearby timepoints $(t, l, s)$.
However, this formulation does not explicitly enforce
$\partial_s f_\theta(x_t, t, s) = v(f_\theta(x_t, t, s), s),$
the defining property that would make $f_\theta$ an integral curve of the true ODE.
Thus, even if Eq. (6) provides sufficient conditions for $f_\theta = f$,
those conditions are not directly constrained in practice, leaving a theoretical gap between local training objectives and the desired global solution behavior.

Moreover, under Classifier-Free Guidance (CFG), SoFlow supervises a guided trajectory
$f_\theta(x_t, t, s; c, w),$
implicitly assuming that
$\partial_s f_\theta(x_t, t, s; c, w) =v_g\big(f_\theta(x_t, t, s; c, w), s \mid c\big), \quad v_g(x, t \mid c) = wv(x, t \mid c) + (1-w)v(x, t),$
where the unconditional component $v(x, t)$ is itself an approximation.
In MeanFlow, this guided field $v_g$ remains a local and well-defined quantity.
In contrast, SoFlow assumes that the integrated, extrapolated trajectory is itself consistent with some valid conditional ODE solution—an assumption that is neither enforced by the objective nor theoretically guaranteed.

From a mathematical standpoint, the learning problem shifts from a well-posed local field approximation
to an ill-posed operator regression problem over nonlinear solution trajectories.
The paper would benefit from clarifying whether the learned $f_\theta$
is empirically observed to satisfy
$
\partial_s f_\theta(x_t, t, s; c, w)
\approx
v_g(f_\theta(x_t, t, s; c, w), s \mid c),
$
and whether such trajectories can be interpreted as genuine integral curves of a consistent guided flow field,
rather than heuristic extrapolations fitted by the loss.


**[W3] Limited evaluation of model performance and validity:**

  The paper reports only FID scores, which primarily reflect perceptual quality but not distributional coverage or faithfulness to the learned dynamics. To properly assess whether the proposed solution operator $f_\theta$  approximates the true ODE solution and preserves the target data distribution, additional quantitative evaluations would be valuable — for instance:
  - Inception Score (IS) or precision/recall to measure coverage and diversity.
  - Deviation or residual norms (e.g., $\| \partial_s f_\theta - v(f_\theta, s)\|$) to measure how closely the learned operator satisfies the underlying ODE.

 Such analyses would better demonstrate whether SoFlow learns a genuinely faithful solution operator rather than merely optimizing perceptual similarity.

**[W4] Explanation for scheduling and mixing effects:**

  While Table 1(a)–(f) includes results for the $k$ scheduling (Eq. 16) and the velocity mix ratio $ m $, the paper provides only empirical trends without clarifying the underlying mechanism.
  For instance, the paper states that a smaller $ m $ improves performance by reducing the randomness of the velocity term, but it is unclear whether this effect arises from variance reduction, regularization, or an implicit bias toward model-predicted guidance.
  Similarly, the scheduling function $ r(k, K) $ in Eq. (16) is varied (exponential, cosine, linear), yet the text does not explain how the schedule theoretically affects optimization stability or convergence.

Could the authors elaborate on why these heuristics influence performance — and whether the model remains stable and consistent under different schedules or $ m $ values?

**[W5] Comparsion table:**
  While the proposed method focuses on one-step generation and demonstrates consistent improvements over MeanFlow across model sizes, the experimental comparison remains somewhat narrow.
  For a fairer assessment of one-step efficiency and quality, it would be beneficial to include either (i) 2-step results of recent consistency-based models or (ii) one-step flow distillation models such as [1~5].
  Including these comparisons would allow readers to more accurately contextualize SoFlow’s efficiency–quality trade-off against both direct training and distilled few-step baselines.

[1] sCT, Simplifying, Stabilizing, and Scaling Continuous-Time Consistency Models., arxiv, 2024\
[2] CTM, Consistency Trajectory Models: Learning Probability Flow ODE Trajectory of Diffusion, ICLR, 2024\
[3] IMM, Inductive Moment Matching, ICML 2025.\
[4] Align Your Flow: Scaling Continuous-Time Flow Map Distillation, arxiv, 2025\
[5] Progressive Distillation, Progressive Distillation for Fast Sampling of Diffusion Models, Neurips, 2023

**Questions:**

Please refer to the “Weaknesses” section for detailed points.

---

> ### Author Response · Authors · 2025-11-28
>
> > [W1] Lack of theoretical grounding in global solution learning
> >
> > SoFlow claims to learn the global ODE solution operator (Eq. (6)) directly, but its training relies only on local consistency objectives (eq. (14)). While this condition theoretically implies $f\_\theta=f$
> > , the actual supervision—based on first-order Taylor approximations and stochastic sampling of $(t,l,s)$
> > —remains inherently local, creating a gap between the theoretical formulation and practical optimization.
> >
> > Traditional Flow Matching constructs a global velocity field via CFM, where local supervision is provably consistent with the true global dynamics. Similarly, Consistency Models enforce locally consistent trajectories using Jacobian-based constraints to form a globally coherent velocity field. In contrast, SoFlow extends this idea to the solution trajectory itself, but without a comparable theoretical foundation.
> >
> > From a functional approximation perspective, to the best of my knowledge, there is no established theoretical evidence that a neural parametrization of the ODE solution trajectory can recover global behavior solely through such locally defined objectives. As this forms the paper’s main conceptual contribution, providing additional theoretical analysis or empirical validation would help substantiate the claim.
>
> We thank the reviewer for pointing out the need for stronger theoretical grounding. **Below, we provide additional theoretical analysis to demonstrate that our model can effectively learn the ground truth solution function based on the local consistency objectives.**
>
> We analyze the error between our model $f\_\theta(x\_t,t,s)$ and the ground truth $f(x\_t,t,s)$. Using the fundamental theorem of calculus, the error can be expressed as:
> $$
>     f(x\_t,t,s)-f\_\theta(x\_t,t,s)=\int\_t^s \frac{\partial }{\partial l}(f\_\theta(f(x\_t,t,l),l,s)dl. \tag{1}
> $$
>
> This equality holds because the boundary terms satisfy $f\_\theta(f(x\_t,t,t),t,s)=f\_\theta(x\_t,t,s)$ and $f\_\theta(f(x\_t,t,s),s,s)=f(x\_t,t,s)$.
>
> Applying the chain rule, the integrand term can be expanded as:
> $$
>       \frac{\partial }{\partial l}(f\_\theta(f(x\_t,t,l),l,s)=\partial\_1 f\_\theta(f(x\_t,t,l),l,s)\partial\_3f(x\_t,t,l)+\partial\_2 f\_\theta(f(x\_t,t,l),l,s). \tag{2}
> $$
>
> Since $f(x\_t,t,s)$ is the ground-truth solution function of the velocity ODE defined by $v(x\_t,t)$, we have $\partial\_3f(x\_t,t,l)=v(f(x\_t,t,l),l)$. We define the residual term of the PDE error as:
> $$
>     R\_\theta(x\_t,t,s) = \partial\_1 f\_\theta(x\_t,t,s)v(x\_t,t)+\partial\_2f\_\theta(x\_t,t,s). \tag{3}
> $$
>
> Substituting this into Eq. (2), we simplify the expression to:
> $$
>         \frac{\partial }{\partial l}(f\_\theta(f(x\_t,t,l),l,s)=R\_\theta(f(x\_t,t,l),l,s). \tag{4}
> $$
>
> Combining this with Eq. (1), we obtain:
> $$
>      f(x\_t,t,s)-f\_\theta(x\_t,t,s)=\int\_t^s R\_\theta(f(x\_t,t,l),l,s)dl. \tag{5}
> $$
>
> During the training process, our model minimizes the difference loss $ \||{f}\_{{{\theta}}}({x}\_t,t,s) -{f}\_{{{\theta}}^-}({x}\_t+v(x\_t,t)(l-t),l,s)\||\_2$ with an adaptive scaling function. To derive an upper bound for the error between our model's prediction and the ground truth, we assume that after training, the loss term $ \||{f}\_{{{\theta}}}({x}\_t,t,s) -{f}\_{{{\theta}}^-}({x}\_t+v(x\_t,t)(l-t),l,s)\||\_2$ is uniformly bounded by $e\_{\text{max}}\|t-l\|$, where $e\_{\text{max}}$ is a constant and $\|t-l\|$ represents the magnitude of the time step.
>
> Assuming $f\_\theta(x\_t,t,s)$ and $v(x\_t,t)$ are twice-continuously differentiable with bounded second-order derivatives, we can apply Taylor's Expansion to bound the discrepancy:
> $$
>     \||({f}\_{{{\theta}}}({x}\_t,t,s)
> -{f}\_{{{\theta}}^-}({x}\_t+v(x\_t,t)(l-t),l,s))-
>     R\_\theta(x\_t,t,s)(t-l)\||\_2 \leq H(t-l)^2. \tag{6}
> $$
>
> This implies the remainder is uniformly bounded by $H(t-l)^2$ given the bounded second-order derivatives. Using the triangle inequality, we combine these bounds:
> $$
>     \||R\_\theta(x\_t,t,s)\||\_2\leq e\_{\text{max}}+H\|t-l\|=e\_{\text{max}}+H\|s-t\|r(k,K), \tag{7}
> $$
>
> where $r(k,K)=\frac{l-t}{s-t}$ is the decreasing schedule function determining $l$ as mentioned in our paper, and $k, K$ denote the current and total training steps, respectively. For simplicity, we denote this upper bound as $\delta$, i.e., $\||R\_\theta(x\_t,t,s)\||\_2\leq\delta$.
>
> Finally, substituting this back into Eq. (5), we arrive at the global error bound:
> $$
>     \||f(x\_t,t,s)-f\_\theta(x\_t,t,s)\||\_2\leq\int\_t^s\||R\_\theta(f(x\_t,t,l),l,s)\||\_2dl\leq\|s-t\|\delta. \tag{8}
> $$
>
> **As the training loss upper bound $e\_{\text{max}}$ and the schedule function value $r(k,K)$ decrease sufficiently during training, our model $f\_\theta(x\_t,t,s)$ effectively approximates the ground truth solution function $f(x\_t,t,s)$ with low error.**

---

> > ### Author Response · Authors · 2025-11-28
> >
> > > [W2] Ambiguity in guidance supervision
> > >
> > > Unlike MeanFlow, which learns a local velocity field $v\_\theta(x, t) \approx \mathbb{E}\_{p(x\_0,x\_1\|x\_t)}[\alpha'\_t x\_0 + \beta'\_t x\_1]$ and guarantees global trajectory consistency via explicit numerical integration, SoFlow directly parameterizes a solution operator $f\_\theta(x\_t, t, s) \approx x\_t + \int\_t^s v(f(x\_t, t, u), u) du$.
> > >
> > > This global mapping is nonlocal—it depends on the entire integral path of $v$—and therefore the training problem is fundamentally different in nature.
> > >
> > > The proposed solution consistency loss (Eq. 14) is based on a first-order Taylor approximation: $f\_\theta(x\_t, t, s) - f\_\theta(x\_t + v(x\_t, t)(l - t), l, s) \propto \partial\_1 f\_\theta(x\_t, t, s)v(x\_t, t) + \partial\_2 f\_\theta(x\_t, t, s)$, which supervises local temporal consistency between nearby timepoints $(t, l, s)$. However, this formulation does not explicitly enforce $\partial\_s f\_\theta(x\_t, t, s) = v(f\_\theta(x\_t, t, s), s)$, the defining property that would make $f\_\theta$ an integral curve of the true ODE. Thus, even if Eq. (6) provides sufficient conditions for $f\_\theta = f$, those conditions are not directly constrained in practice, leaving a theoretical gap between local training objectives and the desired global solution behavior.
> > >
> > > Moreover, under Classifier-Free Guidance (CFG), SoFlow supervises a guided trajectory $f\_\theta(x\_t, t, s; c, w)$, implicitly assuming that $\partial\_s f\_\theta(x\_t, t, s; c, w) = v\_g(f\_\theta(x\_t, t, s; c, w), s \mid c)$,
> > > $$
> > > v\_g(x, t \mid c) = w v(x, t \mid c) + (1 - w)v(x, t)
> > > $$
> > > where the unconditional component $v(x, t)$ is itself an approximation. In MeanFlow, this guided field $v\_g$ remains a local and well-defined quantity. In contrast, SoFlow assumes that the integrated, extrapolated trajectory is itself consistent with some valid conditional ODE solution—an assumption that is neither enforced by the objective nor theoretically guaranteed.
> > >
> > > From a mathematical standpoint, the learning problem shifts from a well-posed local field approximation to an ill-posed operator regression problem over nonlinear solution trajectories. The paper would benefit from clarifying whether the learned $f\_\theta$ is empirically observed to satisfy $\partial\_s f\_\theta(x\_t, t, s; c, w) \approx v\_g(f\_\theta(x\_t, t, s; c, w), s \mid c)$, and whether such trajectories can be interpreted as genuine integral curves of a consistent guided flow field, rather than heuristic extrapolations fitted by the loss.
> >
> > We appreciate the reviewer's insight regarding the theoretical gap between local training objectives and the desired global solution behavior, as well as the questions regarding whether the learned $f\_\theta$ satisfies the ODE condition $\partial\_3f\_\theta(x\_t,t,s,c)=v\_g(f\_\theta(x\_t,t,s,c),s,c)$ for CFG training. Here, we treat $w$ as a constant. We provide detailed explanations for these two points below:
> >
> > 1. **We provide theorectical proof to demonstrate that our local training objectives indeed enforce the desired global solution behavior implicitly.** We follow the notations introduced in W1.
> >
> > To derive a bound for the ODE error, we make three mild assumptions. First, we assume that after training, the residual satisfies $\||R\_\theta(x\_t,t,s)\||\_2 \leq \delta$ for all $x \in \mathbb{R}^n$ and $0 \leq s \leq t \leq 1$, where $\delta=e\_{\text{max}}+H\|s-t\|r(k,K)$ as mentioned in W1.
> >
> > Second, we assume that $v(x\_t,t)$ is continuously differentiable and Lipschitz continuous with respect to $x\_t$, i.e., $\||v(x\_t,t)-v(y\_t,t)\||\_2\leq K\||x\_t-y\_t\||\_2$.
> >
> > Third, we assume that $R\_\theta(x\_t,t,s)$ is continuously differentiable and its partial derivative with respect to $s$ is Lipschitz continuous; that is, $\|| \partial\_3 R\_\theta(x\_t,t,s\_1) - \partial\_3 R\_\theta(x\_t,t,s\_2) \||\_2 \leq L\|s\_1 - s\_2\|$, where $L$ is the Lipschitz constant.
> >
> > We first prove the following lemma:
> >
> > **Lemma**: Under the stated assumptions, $\partial\_3 R(x\_t,t,s)$ is uniformly bounded by:
> > $$
> >     \||\partial\_3 R\_\theta(x\_t,t,s)\||\_2 \leq 2\sqrt{L\delta}. \tag{9}
> > $$

---

> > > ### Author Response · Authors · 2025-11-28
> > >
> > > Proof: For brevity, we denote $R\_\theta(x\_t, t, s)$ as $g(s)\in \mathbb{R}^n$. We aim to bound $\||g'(s)\||\_2$ given that $\||g(s)\||\_2 \leq \delta$ and $\||g'(s\_1) - g'(s\_2)\||\_2 \leq L\|s\_1 - s\_2\|$.
> > > According to the fundamental theorem of calculus, we have:
> > > $$
> > >     g(s+h)-g(s)=\int\_0^1g^\prime(s+\tau h)hd\tau. \tag{10}
> > > $$
> > >
> > > We can rewrite the term $g^\prime(s)h$ as follows:
> > > $$
> > > \begin{aligned}
> > >         g^\prime(s)h&= g^\prime(s)h+ g(s+h)-g(s)-\int\_0^1g^\prime(s+\tau h)hd\tau\\\\
> > >         &= g(s+h)-g(s) - \int\_0^1(g^\prime(s+\tau h)-g^\prime(s))hd\tau.
> > > \end{aligned} \tag{11}
> > > $$
> > >
> > > Taking the Euclidean norm on both sides and applying the triangle inequality yields:
> > > $$
> > >     \|h\| \||g'(s)\||\_2 \leq \||g(s+h)\||\_2 + \||g(s)\||\_2 + \left\|| \int\_{0}^{1} (g'(s+\tau h) - g'(s)) h \, d\tau \right\||\_2. \tag{12}
> > > $$
> > >
> > > Using the assumption $\||g(\cdot)\||\_2 \leq \delta$ and the Lipschitz continuity of $g'(s)$, we can bound the terms using the triangle inequality for integrals:
> > > $$
> > > \begin{aligned}
> > >     \|h\| \||g'(s)\||\_2 &\leq \delta + \delta + \int\_{0}^{1} \||g'(s+\tau h) - g'(s)\||\_2 \|h\| \, d\tau \\\\
> > >     &\leq 2\delta + \int\_{0}^{1} L \|\tau h\| \|h\| \, d\tau \\\\
> > >     &= 2\delta + L h^2 \int\_{0}^{1} \tau \, d\tau \\\\
> > >     &= 2\delta + \frac{L h^2}{2}.
> > > \end{aligned} \tag{13}
> > > $$
> > >
> > > Note that this inequality holds for any $h$. We consider the case where $h>0$ to form an upper bound. Dividing by $\|h\|$, we obtain:
> > > $$
> > >     \||g'(s)\||\_2 \leq \frac{2\delta}{h} + \frac{Lh}{2}, \quad \forall h>0. \tag{14}
> > > $$
> > >
> > > To find the tightest bound, we minimize the right-hand side with respect to $h$. The minimum occurs when the derivative with respect to $h$ is zero (since the second derivative is positive), yielding:
> > > $$
> > >     -\frac{2\delta}{h^2} + \frac{L}{2} = 0 \implies h^2 = \frac{4\delta}{L} \implies h = 2\sqrt{\frac{\delta}{L}}. \tag{15}
> > > $$
> > >
> > > Substituting this optimal $h$ back into the inequality gives:
> > > $$
> > >     \||g'(s)\||\_2 \leq \frac{2\delta}{2\sqrt{\delta/L}} + \frac{L}{2} \left(2\sqrt{\frac{\delta}{L}}\right) = \sqrt{L\delta} + \sqrt{L\delta} = 2\sqrt{L\delta}. \tag{16}
> > > $$
> > > This concludes the proof of the lemma.
> > >
> > > Next, taking the partial derivatives with respect to $s$ in Eq. (5), we have:
> > > $$
> > >     \partial\_3 f(x\_t,t,s)-\partial\_3 f\_\theta(x\_t,t,s)=R\_\theta(f(x\_t,t,s),s,s)+\int\_t^s\partial\_3R\_\theta(f(x\_t,t,l),l,s)dl. \tag{17}
> > > $$
> > >
> > > Using this equation, we can write the ODE residual vector as follows:
> > > $$
> > > \begin{aligned}
> > >     &\partial\_3 f\_\theta(x\_t,t,s)-v(f\_\theta(x\_t,t,s),s)\\\\=\partial\_3 f(x\_t,t,s)-v(f\_\theta(x\_t&,t,s),s)-R\_\theta(f(x\_t,t,s),s,s)-\int\_t^s\partial\_3R\_\theta(f(x\_t,t,l),l,s)dl\\\\
> > >     =v(f(x\_t,t,s),s)-v(f\_\theta&(x\_t,t,s),s)-R\_\theta(f(x\_t,t,s),s,s)-\int\_t^s\partial\_3R\_\theta(f(x\_t,t,l),l,s)dl.
> > > \end{aligned} \tag{18}
> > > $$
> > >
> > > We first use the Lipschitz continuity of $v(x\_t,t)$, along with Eq. (8):
> > > $$
> > >     \||v(f(x\_t,t,s),s)-v(f\_\theta(x\_t,t,s),s)\||\_2\leq K\||f(x\_t,t,s)-f\_\theta(x\_t,t,s)\||\_2\leq K\|s-t\|\delta. \tag{19}
> > > $$
> > >
> > > Since we have a uniform bound $\||R\_\theta(x\_t,t,s)\||\_2\leq \delta$, it follows that:
> > > $$
> > >     \||R\_\theta(f(x\_t,t,s),s,s)\||\_2\leq\delta. \tag{20}
> > > $$
> > >
> > > Finally, for the integral term, using the triangle inequality and the lemma, we have:
> > > $$
> > > \begin{aligned}
> > >     \left\||\int\_t^s\partial\_3R\_\theta(f(x\_t,t,l),l,s)dl\right\||\_2&\leq \left|\int\_t^s\||\partial\_3R\_\theta(f(x\_t,t,l),l,s)\||\_2dl\right|\\\\&\leq\left|\int\_s^t 2\sqrt{L\delta}dl\right|=2\|s-t\|\sqrt{L\delta}.
> > > \end{aligned} \tag{21}
> > > $$
> > >
> > > Combining these three terms, we obtain:
> > > $$
> > >     \||\partial\_3 f\_\theta(x\_t,t,s)-v(f\_\theta(x\_t,t,s),s)\||\_2\leq K\|s-t\|\delta+\delta+2\|s-t\|\sqrt{L\delta}=O(\sqrt{\delta}). \tag{22}
> > > $$
> > >
> > > We have theoretically proved that the ODE error $\|| \partial\_3 f\_\theta(x\_t,t,s)-v(f\_\theta(x\_t,t,s),s) \||\_2$ of our model $f\_\theta(x\_t,t,s)$ is bounded by the PDE error $\||R\_\theta(x\_t,t,s)\||\_2$. From the response to W1, we know $\delta=e\_{\text{max}}+H\|s-t\|r(k,K)$. **During training, the error $e\_{\text{max}}$ is minimized and the schedule function $r(k,K)$ also decreases to a small value. Thus, we conclude that our model implicitly minimizes the ODE error during training with a theoretical guarantee.**

---

> ### Author Response · Authors · 2025-11-28
>
> 2. We have revised the CFG section of our paper to provide better clarification. **Here, provide theorectical proof to demonstrate that our CFG training objectives indeed enforce $\partial_s f_\theta(x_t, t, s, c) \approx v_g(f_\theta(x_t, t, s, c), s \mid c)$ in an implicit manner.** First, we define the guided marginal velocity field $v\_g(x\_t,t,s)$ as the linear combination of conditional and unconditional marginal velocity fields:
> $$
> \begin{aligned}
> {v}\_g({x}\_t, t, {c}) &= w {v}({x}\_t, t \mid {c}) + (1-w) {v}({x}\_t, t)\\\\
> =w\mathbb{E}\_{p(x\_1,x\_0|x\_t,c)}[\alpha\_t^\prime x\_0&+\beta\_t^\prime x\_1]+(1-w)\mathbb{E}\_{p(x\_1,x\_0|x\_t)}[\alpha\_t^\prime x\_0+\beta\_t^\prime x\_1].
> \end{aligned} \tag{23}
> $$
>
> This guided field is well-defined by the data distribution and is independent of the model. We denote $f(x\_t,t,s,c)$ as the ground-truth solution function to the velocity ODE defined by $v\_g(x\_t,t,c)$. Analogous to the unconditional setting, we require our model $f\_\theta(x,t,s,c)$ to satisfy the following two equations to ensure it learns $f(x\_t,t,s,c)$ for every condition $c$:
> $$
> \begin{align}
> &f\_\theta(x,t,t,c)=x\_t, \tag{24}\\\\
> &\partial\_1f\_\theta(x\_t,t,s,c)v\_g(x\_t,t,c)+\partial\_2f\_\theta(x\_t,t,s,c)=0. \tag{25}
> \end{align}
> $$
>
> To achieve this goal, we define a guided flow matching loss $ L\_{\mathrm{FM}}^g({{\theta}})$ as:
> $$
>     \mathbb{E}\_{t,{x}\_t,c}\left[ \frac{w\_{\mathrm{FM}}(t,\mathrm{MSE})}{D}\left\|| \partial\_2a(t,t){x}\_t+\partial\_2b(t,t){F}\_{{\theta}}({x}\_t,t,t,c)- v\_g(x\_t,t,c)\right\||\_2^2 \right], \tag{26}
> $$
>
> and a guided solution consistency loss $L\_{\mathrm{SCM}}^g({{\theta}})$ as:
> $$
>     \mathbb{E}\_{t,l,s,{x}\_t,c}
>     \left[
>         \frac{w\_{\mathrm{SCM}}(t,l,s,\mathrm{MSE})}{D}
>         \left\||
>             {f}\_{{{\theta}}}({x}\_t,t,s,c)
>             -{f}\_{{{\theta}}^-}\bigl({x}\_t+v\_g(x\_t,t,c)(l-t),l,s,c\bigr)
>         \right\||\_2^2
>     \right], \tag{27}
> $$
>
> where $D$ is the data dimension, $\theta^-$ denotes the stop-gradient operator applied to the targets, and the adaptive weighting functions remain consistent with the unconditional case. The total guided training loss $L^g({{\theta}})$ is defined as their linear combination $\lambda L\_{\mathrm{FM}}^g({{\theta}})+(1-\lambda)L\_{\mathrm{SCM}}^g({{\theta}})$.
>
> We clarify that, since the unconditional term $v(x\_t,t)$ in $v\_g(x\_t,t,c)$ is not directly accessible in conditional generation settings, we use our model's prediction to provide an estimation of $v(x\_t,t)$. However, this does not imply that the ground-truth guided field relies on a specific model.
> **The ground-truth guided field $v\_g(x\_t,t,c)$ and the corresponding solution function $f(x\_t,t,s,c)$ are determined solely by data distributions.**
>
> Regarding the upper bound of $\|| \partial\_3f\_\theta(x\_t,t,s,c)-v\_g(f\_\theta(x\_t,t,s),s,c) \||\_2$, we first consider:
> $$
>     \||{f}\_{{{\theta}}}({x}\_t,t,s,c)
> -{f}\_{{{\theta}}^-}({x}\_t+v\_g(x\_t,t,c)(l-t),l,s,c)\||\_2. \tag{28}
> $$
>
> During training, we use the following mixed guided velocity field $v\_\text{mix}$ to replace the intractable guided field $v\_g(x\_t,t,c)$ to provide a tractable target:
> $$
>     v\_\text{mix} = m(wv(x\_t,t\|c)+(1-w)\partial\_3f\_\theta(x\_t,t,t,\phi))+(1-m)\partial\_3f\_\theta(x\_t,t,t,c), \tag{29}
> $$
>
> where the conditional marginal velocity field $v(x\_t,t\|c)=\mathbb{E}\_{p(x\_1,x\_0\|x\_t,c)}[\alpha\_t^\prime x\_0+\beta\_t^\prime x\_1]$ can be directly replaced with a random estimation during training, similar to the standard flow matching framework. We assume that after training,
> $$
>     \||{f}\_{{{\theta}}}({x}\_t,t,s,c)
> -{f}\_{{{\theta}}^-}({x}\_t+v\_\text{mix}(x\_t,t,c)(l-t),l,s,c)\||\_2\leq e\_{\text{mix}}\|t-l\|, \tag{30}
> $$
>
> which is similar to the unconditional situation. Furthermore, since the unconditional flow matching loss and guided flow matching loss optimize our model to predict the unconditional and guided marginal velocity fields, we can also assume that after training these losses are bounded:
> $$
> \begin{align}
>     &\||\partial\_3f\_\theta(x\_t,t,t,\phi)-v(x\_t,t)\||\_2\leq e\_{\text{uncond}}, \tag{31}\\\\
>     &\||\partial\_3f\_\theta(x\_t,t,t,c)-v\_g(x\_t,t,c)\||\_2\leq e\_{\text{guided}}. \tag{32}
> \end{align}
> $$

---

> > ### Author Response · Authors · 2025-11-28
> >
> > According to the following equation:
> > $$
> > \begin{aligned}
> >     v\_\text{mix}(x\_t,t,c)-&v\_g(x\_t,t,c) = \\\\m(1-w)(\partial\_3f\_\theta(x\_t,t,t,\phi)-v(x\_t,t))+&(1-m) (\partial\_3f\_\theta(x\_t,t,t,c)-v\_g(x\_t,t,c)),
> > \end{aligned} \tag{33}
> > $$
> >
> > we know $v\_\text{mix}(x\_t,t,c)$ is also close to $v\_g(x\_t,t,c)$ since:
> > $$
> >     \||v\_\text{mix}(x\_t,t,c)-v\_g(x\_t,t,c)\||\_2\leq m(1-w)e\_\text{uncond}+(1-m)e\_\text{guided}. \tag{34}
> > $$
> >
> > Combining this inequality with Eq. (30), we have:
> > $$
> > \begin{aligned}
> >      \||{f}\_{{{\theta}}}({x}\_t,t,s,c)&
> > -{f}\_{{{\theta}}^-}({x}\_t+v\_g(x\_t,t,c)(l-t),l,s,c)\||\_2\leq e\_\text{max}\|t-l\|,\\\\
> > e\_\text{max}=&e\_\text{mix}+L\_f(m(1-w)e\_\text{uncond}+(1-m)e\_\text{guided}),
> > \end{aligned} \tag{35}
> > $$
> >
> > where we assume $f\_\theta$ is Lipschitz continuous with respect to the first variable and $L\_f$ is the Lipschitz constant. We can then consider the residual:
> > $$
> >     R\_\theta(x\_t,t,s,c) = \partial\_1 f\_\theta(x\_t,t,s,c)v\_g(x\_t,t,c)+\partial\_2f\_\theta(x\_t,t,s,c). \tag{36}
> > $$
> >
> > By assuming $f\_\theta$ is twice-continuously differentiable with bounded second-order derivatives, we obtain results similar to the unconditional setting:
> > $$
> >     \||({f}\_{{{\theta}}}({x}\_t,t,s,c)
> > -{f}\_{{{\theta}}^-}({x}\_t+v\_g(x\_t,t,c)(l-t),l,s,c))-
> >     R\_\theta(x\_t,t,s,c)(t-l)\||\_2 \leq H(t-l)^2, \tag{37}
> > $$
> >
> > where $H$ is a constant related to the upper bound of the second-order derivatives of $f\_\theta$. We can see that if we denote $\delta=e\_\text{max}+H\|t-l\|=e\_\text{max}+H\|s-t\|r(k,K)$ again, we can directly follow the proofs in Part 1 to demonstrate that:
> > $$
> >      \||\partial\_3f\_\theta(x\_t,t,s,c)-v\_g(f\_\theta(x\_t,t,s),s,c)\||\_2\leq O(\sqrt{\delta}), \tag{38}
> > $$
> > **which shows that our guided model theoretically minimizes the ODE error in the CFG training process implicitly, analogous to the unconditional situation.**

---

> ### Author Response · Authors · 2025-11-28
>
> > W3: Limited evaluation of model performance and validity:
> >
> > The paper reports only FID scores, which primarily reflect perceptual quality but not distributional coverage or faithfulness to the learned dynamics. To properly assess whether the proposed solution operator $f\_\theta$ approximates the true ODE solution and preserves the target data distribution, additional quantitative evaluations would be valuable — for instance:
> > *   Inception Score (IS) or precision/recall to measure coverage and diversity.
> > *   Deviation or residual norms (e.g., $\|\partial\_s f\_\theta - v(f\_\theta, s)\|$) to measure how closely the learned operator satisfies the underlying ODE.
> >
> > Such analyses would better demonstrate whether SoFlow learns a genuinely faithful solution operator rather than merely optimizing perceptual similarity.
>
> As demonstrated in our theoretical analysis in W2, the ODE error is implicitly minimized by our training objective in both unconditional and CFG settings. In this section, we provide additional empirical results to assess the model's distributional coverage and diversity.
>
> Regarding the reviewer's suggestion to measure the deviation norm $\|\|\partial\_3f\_\theta(x\_t,t,s,c)-v\_g(f\_\theta(x\_t,t,s),s,c)\|\|\_2$, **we note that this quantity cannot be precisely measured in the CFG setting.** This is because the ground truth guided velocity field:
>
> $$
> v\_g(x\_t,t,c)=  w\mathbb{E}\_{p(x\_1,x\_0\|x\_t,c)}[\alpha\_t^\prime x\_0+\beta\_t^\prime x\_1]+(1-w)\mathbb{E}\_{p(x\_1,x\_0\|x\_t)}[\alpha\_t^\prime x\_0+\beta\_t^\prime x\_1]
> $$
>
> **is a linear combination of intractable posterior expectations.** Therefore, we instead report 1-NFE and 2-NFE generation results using Inception Score (IS), FID, sFID, Precision, and Recall to comprehensively evaluate model performance.
>
> Prior to this evaluation, we revisited our hyperparameter settings. We found that the CFG strength of 3.0, originally determined via ablation on the smaller SoFlow-B/4 model, was suboptimal for our larger-scale models, which generally benefit from lower guidance scales. By optimizing the CFG strength to 2.5, 2.25, 2.0, and 2.0 for SoFlow-B/2, M/2, L/2, and XL/2, respectively, we observed significant performance improvements.
>
> Our updated results (Table 1) demonstrate the effectiveness of this approach. Specifically, for 1-NFE and 2-NFE generation, our SoFlow-XL/2 model achieves FID scores of **2.962** and **2.661**, significantly outperforming the MeanFlow-XL/2 baseline (3.43 and 2.93). **In addition, for IS, sFID, precision and recall metrics, SoFlow-XL/2 shows competitive few-step generation capabilities compared to the DiT-XL/2 model.** This is particularly notable given that our model is trained for only **240** epochs, whereas DiT-XL/2 is trained for **1400** epochs and utilizes **250$\times$2 NFE** for inference.
>
> **Table 1: Additional quantitative 1-NFE and 2-NFE results on generation fidelity and diversity.**
>
> | Model | SoFlow-B/2 | SoFlow-M/2 | SoFlow-L/2 | SoFlow-XL/2 |
> | :--- | :---: | :---: | :---: | :---: |
> | CFG $w$ | 2.5 | 2.25 | 2.0 | 2.0 |
> | Epochs | 240 | 240 | 240 | 240 |
> | **_1-NFE_** | | | | |
> | IS | 218.373 | 235.147 | 235.218 | **238.530** |
> | FID-50K | 4.849 | 3.733 | 3.201 | **2.962** |
> | sFID-50K | 5.395 | 5.356 | 4.865 | **4.600** |
> | Precision | **0.798** | 0.796 | **0.798** | 0.795 |
> | Recall | 0.490 | 0.524 | 0.555 | **0.564** |
> | **_2-NFE_** | | | | |
> | IS | 260.472 | **276.717** | 269.642 | 269.477 |
> | FID-50K | 4.244 | 3.423 | 2.899 | **2.661** |
> | sFID-50K | 4.695 | 4.746 | 4.836 | **4.562** |
> | Precision | **0.864** | 0.854 | 0.840 | 0.835 |
> | Recall | 0.437 | 0.489 | 0.525 | **0.540** |
>
> | Metrics | NFE | CFG $w$ | Epochs | IS | FID | sFID | Precision | Recall |
> | :--- | :---: | :---: | :---: | :---: | :---: | :---: | :---: | :---: |
> | DiT-XL/2 | 250$\times$2 | 1.5 | 1400 | 278.24 | 2.27 | 4.60 | 0.83 | 0.57 |

---

> > ### Author Response · Authors · 2025-11-28
> >
> > > W4: Explanation for scheduling and mixing effects:
> > >
> > > While Table 1(a)–(f) includes results for the $k$ scheduling (Eq. 16) and the velocity mix ratio $m$, the paper provides only empirical trends without clarifying the underlying mechanism. For instance, the paper states that a smaller $m$ improves performance by reducing the randomness of the velocity term, but it is unclear whether this effect arises from variance reduction, regularization, or an implicit bias toward model-predicted guidance. Similarly, the scheduling function $r(k,K)$ in Eq. (16) is varied (exponential, cosine, linear), yet the text does not explain how the schedule theoretically affects optimization stability or convergence.
> > >
> > > Could the authors elaborate on why these heuristics influence performance — and whether the model remains stable and consistent under different schedules or $m$ values?
> >
> > **In this section, we provide theoretical insights into the schedule function $r(k,K)$ and the velocity mix ratio $m$ under the Euler parameterization, as we adopt this formulation in our large-scale experiments.**
> >
> > We first analyze the effect of $r(k,K)$. Under the Euler parameterization:
> >
> > $$
> > \begin{aligned}
> >     &f\_\theta(x\_t,t,s)-f\_{\theta^-}(x\_t+v(x\_t,t)(l-t),l,s) \\\\
> > =&(x\_t+(s-t)F\_\theta(x\_t,t,s)) - (x\_t+v(x\_t,t)(l-t)+(s-l)F\_{\theta^-}(x\_t+v(x\_t,t)(l-t),l,s)) \\\\
> > =&(s-t)\left(F\_\theta(x\_t,t,s)-\left(\frac{l-t}{s-t}v(x\_t,t)+\frac{s-l}{s-t}F\_{\theta^-}(x\_t+v(x\_t,t)(l-t),l,s)\right)\right) \\\\
> > =&(s-t)\left(F\_\theta(x\_t,t,s)-\left(r(k,K)v(x\_t,t)+(1-r(k,K))F\_{\theta^-}(x\_t+v(x\_t,t)(l-t),l,s)\right)\right),
> > \end{aligned}
> > $$
> >
> > where $r(k,K)$ is our schedule function that decreases monotonically from the beginning to the end of training. This derivation indicates that, ignoring weighting coefficients, our loss function is proportional to the mean squared error:
> >
> > $$
> > \mathbb{E}\_{t,l,s,x\_t}\left[\left\|\|F\_\theta(x\_t,t,s)-\left(r(k,K)v(x\_t,t)+(1-r(k,K))F\_{\theta^-}(x\_t+v(x\_t,t)(l-t),l,s)\right)\right\|\|\_2^2\right].
> > $$
> >
> > According to this formula, **$r(k,K)$ controls the interpolation ratio between the instantaneous velocity field and our model's prediction at a slightly lower noise level.** When $r(k,K)=1$, our target degenerates to the flow matching target. As $r(k,K)$ decays during training, the instantaneous velocity component in the target gradually decreases, smoothly guiding the model to learn the solution function with higher approximation accuracy. Thus, different schedules for $r(k,K)$ control how fast the instantaneous velocity component decays, and we empirically find that an exponential schedule performs well.
> >
> > Regarding the velocity mix ratio $m$, we consider the target mentioned above for the guided training setting. Specifically, the target is
> >
> > $$
> > r(k,K)v\_\text{mix}+(1-r(k,K))F\_{\theta^-}(x\_t+v\_\text{mix}(l-t),l,s),
> > $$
> >
> > where the mixed velocity field $v\_\text{mix}$ is defined as
> >
> > $$
> > m(w(\alpha\_t^\prime x\_0+\beta\_t^\prime x\_1)+(1-w)v\_{\text{uncond}})+(1-m)v\_{\text{guide}}.
> > $$
> >
> > Thus, the target is effectively a linear interpolation between $\alpha\_t^\prime x\_0+\beta\_t^\prime x\_1$, $v\_{\text{uncond}}=\partial\_3f\_{\theta^-}(x\_t,t,t,\phi)$, $v\_{\text{guide}}=\partial\_3f\_{\theta^-}(x\_t,t,t,c)$, and $F\_{\theta^-}(x\_t+v\_\text{mix}(l-t),l,s)$, with coefficients $r(k,K)mw$, $r(k,K)m(1-w)$, $r(k,K)(1-m)$, and $(1-r(k,K))$. We can observe that, given $x\_t$, $t$, and $c$, **most of the randomness arises from the term $\alpha\_t^\prime x\_0+\beta\_t^\prime x\_1$**, since $v\_{\text{uncond}}$ and $v\_{\text{guide}}$ are fully determined by $x\_t, t, c$. Furthermore, $F\_{\theta^-}(x\_t+v\_\text{mix}(l-t),l,s)$ involves the term $l-t$, which is small, thereby limiting the propagation of randomness. **Therefore, choosing a small velocity mix ratio $m$ reduces the target's variance by scaling down the stochastic $\alpha\_t^\prime x\_0+\beta\_t^\prime x\_1$ term.**
> >
> > Finally, our ablation studies indicate that the model remains stable and functional under different schedules and $m$ values, although specific performance metrics are affected by these detailed choices.

---

> > > ### Author Response · Authors · 2025-11-28
> > >
> > > > W5: Comparsion table:
> > > >
> > > > While the proposed method focuses on one-step generation and demonstrates consistent improvements over MeanFlow across model sizes, the experimental comparison remains somewhat narrow.
> > > >
> > > > For a fairer assessment of one-step efficiency and quality, it would be beneficial to include either (i) 2-step results of recent consistency-based models or (ii) one-step flow distillation models such as [1-5]. Including these comparisons would allow readers to more accurately contextualize SoFlow’s efficiency–quality trade-off against both direct training and distilled few-step baselines.
> > > >
> > > > [1] sCT, Simplifying, Stabilizing, and Scaling Continuous-Time Consistency Models., arxiv, 2024
> > > >
> > > > [2] CTM, Consistency Trajectory Models: Learning Probability Flow ODE Trajectory of Diffusion, ICLR, 2024
> > > >
> > > > [3] IMM, Inductive Moment Matching, ICML 2025.
> > > >
> > > > [4] Align Your Flow: Scaling Continuous-Time Flow Map Distillation, arxiv, 2025
> > > >
> > > > [5] Progressive Distillation, Progressive Distillation for Fast Sampling of Diffusion Models, Neurips, 2023
> > >
> > > Regarding the comparison with the suggested baselines, we note that direct comparisons with sCT [1] and Align Your Flow [4] are not feasible, as they report results on **ImageNet-512$\times$512**. Similarly, CTM [2] and Progressive Distillation [5] focus on **ImageNet-64$\times$64**. In contrast, our models are trained and evaluated on **ImageNet-256$\times$256**.
> > >
> > > Instead, we compare our 1-step and 2-step results with iCT, IMM, Shortcut Models, and MeanFlow, which operate under comparable settings. Using the optimized settings detailed in W3, SoFlow-XL/2 achieves a 1-NFE FID-50K of **2.962** and a 2-NFE FID-50K of **2.661** after 240 epochs. These scores significantly outperform MeanFlow-XL/2, which achieves 3.43 (1-NFE) and 2.93 (2-NFE).
> > >
> > > **Furthermore, our smaller models consistently surpass larger MeanFlow counterparts.** Specifically, for 1-NFE generation, MeanFlow reports FID-50K values of 5.01 (M), 3.84 (L), and 3.43 (XL). In comparison, our optimized SoFlow-B, M, and L models achieve superior scores of 4.849, 3.733, and 3.201, respectively. These results demonstrate that our method delivers strong few-step performance and maintains an excellent trade-off between parameter efficiency and generation quality.
> > >
> > > **Table 2: Comparison of 1-NFE and 2-NFE generation results on ImageNet-256$\times$256 dataset.**
> > >
> > > | method | epochs | params | NFE | FID-50K $\downarrow$ |
> > > | :--- | :---: | :---: | :---: | :---: |
> > > | iCT-XL / 2 | - | 675M | 1 / 2 | 34.24 / 20.30 |
> > > | Shortcut-XL / 2 | 250 | 675M | 1 / 4 | 10.60 / 7.8 |
> > > | IMM-XL / 2 | 3840 | 675M | 1$\times$2 / 2$\times$2 | 7.77 / 3.99 |
> > > | MeanFlow-B / 2 | 240 | 131M | 1 | 6.17 |
> > > | MeanFlow-M / 2 | 240 | 308M | 1 | 5.01 |
> > > | MeanFlow-L / 2 | 240 | 459M | 1 | 3.84 |
> > > | MeanFlow-XL / 2 | 240 | 675M | 1 / 2 | 3.43 / 2.93 |
> > > | SoFlow-B / 2 | 240 | 131M | 1 / 2 | 4.849 / 4.244 |
> > > | SoFlow-M / 2 | 240 | 308M | 1 / 2 | 3.733 / 3.423 |
> > > | SoFlow-L / 2 | 240 | 459M | 1 / 2 | 3.201 / 2.899 |
> > > | SoFlow-XL / 2 | 240 | 675M | 1 / 2 | **2.962** / **2.661** |

---

### Official Review · Reviewer_aKZH · 2025-10-22

**Soundness:** 3
**Presentation:** 1
**Contribution:** 4
**Rating:** 6
**Confidence:** 4

**Summary:**

This paper proposed the SoFlow, a one-step generation paradigm for flow-matching diffusion models (FM). The main motivation for this paper is to directly approximate the unique solution of the ODE of the FM. The experimental results show that SoFlow could successfully achieve the one-step generation.

**Strengths:**

1. This paper proposes a novel method for one-step generation. This paper demonstrates how to approximate the unique solution of the ODE for the FM using sound theoretical proofs. Then, they build a learn target to train the model. In this way, one-step generation could be regarded as the solution from T to 0, thereby being directly solved via SoFlow. To the best of my knowledge, this method is the first one to achieve this and will truly benefit the development of FM.

2. The experimental results clearly show that SoFlow works well.

**Weaknesses:**

1. I have to say that the writing of this paper should undergo a significant revision. Firstly, in Eq. 1, there is no explanation for $\alpha^{'}_t$ and $b^{'}_t$. Eq 5 should at least be split into two equations, not integrated into one. I highly recommend that the author add the integrated formulation of a unique solution. In this way, the reader could clearly know why $f(x_t, t, t)$ contains two time-related variables, one for the start and one for the end. Meanwhile, it may be better to expand Eq. 6 - Eq. 12 in the Appendix. This section is one of the most crucial parts of this paper and warrants a more detailed explanation. Meanwhile, this could simplify the audience's budget. Then, why is there no equation number in lines 192 and 202 lines.  The equation in 192 lines should also explain how it is derived. This is also important and directly related to your loss. Then, $\alpha$ and $b$ are redefined twice in 123 lines and again in Eq. 10, which may mislead the reader. In the end, the CFG section should reorder the logic. Currently, it is challenging for the reader to understand the rationale behind introducing a condition. The solution of the ODE itself does not contain the condition $c$. Meanwhile, the dataset itself cannot recognize the condition. This will lead the reader to question the validity of the CFG section.

2. I think the author should add a section to illustrate how to achieve distillation. Well, training from scratch is necessary, but distillation is also essential and can further enhance the contribution of this paper (can refer to the consistency models).

3. Experimental results in the Table. 1 has some inconsistency in the Table. 2. For example, the noise scheduler part shows the best FID is 59.97. But the true best FID is 3.35.

**Questions:**

1. How much influence does using the expectation of the velocity to replacing the velocity term? Is there any way to quantify the bias after replacing it with expectation?

To summarize, I believe this paper is novel. Approximating the ODE solution directly is an interesting approach. The experimental results prove its validity. But the writing should be clearer. The primary concern for me is the writing of this paper, which I believe does not meet the standards of ICLR. During the rebuttal, I recommend that the author improve the quality of their writing. Therefore, I rate it as marginally above the acceptance threshold, temporarily.

---

> ### Author Response · Authors · 2025-11-28
>
> > W1: I have to say that the writing of this paper should undergo a significant revision. Firstly, in Eq. 1, there is no explanation for
> $\alpha\_t^\prime$ and $b\_t^\prime$. Eq 5 should at least be split into two equations, not integrated into one. I highly recommend that the author add the integrated formulation of a unique solution. In this way, the reader could clearly know why $f(x\_t,t,t)$ contains two time-related variables, one for the start and one for the end. Meanwhile, it may be better to expand Eq. 6 - Eq. 12 in the Appendix. This section is one of the most crucial parts of this paper and warrants a more detailed explanation. Meanwhile, this could simplify the audience's budget. Then, why is there no equation number in lines 192 and 202 lines. The equation in 192 lines should also explain how it is derived. This is also important and directly related to your loss. Then,
>  $\alpha$ and $b$ are redefined twice in 123 lines and again in Eq. 10, which may mislead the reader. In the end, the CFG section should reorder the logic. Currently, it is challenging for the reader to understand the rationale behind introducing a condition. The solution of the ODE itself does not contain the condition $c$. Meanwhile, the dataset itself cannot recognize the condition. This will lead the reader to question the validity of the CFG section.
>
> We thank the reviewer for their constructive feedback on improving the clarity and presentation of our paper. **We have carefully addressed all the raised points in our revised manuscript to provide a better understanding.**
>
> 1. In response to the comment on Eq. 1, we have now included explicit definitions for $\alpha\_t^\prime$ and $\beta\_t^\prime$ as the derivatives of $\alpha\_t$ and $\beta\_t$, respectively.
>
> 2. Following the suggestion, we have split the original Eq. 5 into two separate equations and added the integrated formulation in the new Eq. 7 to enhance clarity.
>
> 3. Regarding Eqs. 6--12, we believe these equations are self-contained, as they rely primarily on fundamental derivative properties. Specifically, Eq. 6 contains the conditions that our model should satisfy, Eqs. 7--8 utilize the chain rule and the implication of zero derivatives to establish the sufficiency of our proposed condition for learning the solution function. Eqs. 9--10 apply basic derivative rules to derive the flow matching loss, Eq. 11 presents the final form of this loss, and Eq. 12 introduces the adaptive weighting function. If the reviewer have further specific concerns, we would be glad to provide additional clarification.
>
> 4. We have added equation numbers to the equations originally located at lines 192 and 202. We had initially omitted these numbers under the assumption that they pertained to parameterization choices rather than mathematical derivations.
>
> 5. We would like to clarify a potential misunderstanding regarding the notation. We strictly distinguish between two sets of variables:
>     *   In Line 123, we define the **Greek letters** $\alpha\_t$ and $\beta\_t$ as single-variable functions for the flow matching trajectory.
>     *   In Eq. 10, we utilize the **Latin letters** $a(t,s)$ and $b(t,s)$ as two-variable functions for the model's parameterization.
>
>     The reviewer mentioned that "$\alpha$ and $b$" are redefined. Since $\alpha$ is paired with $\beta$, and $a$ is paired with $b$, we suspect there might be a confusion between these distinct symbols.
>
> 6. **We have thoroughly restructured the CFG section to improve logical flow and readability.** In particular, we now provide a detailed explanation of how the condition $c$ is incorporated into our framework. Under conditional generation, the model $f\_{\theta}(x\_t,t,s,c)$ is designed to approximate the solution function of the ODE defined by the guided marginal velocity field:
> $$
> {v}\_g({x}\_t, t, {c}) = w {v}({x}\_t, t \mid {c}) + (1-w) {v}({x}\_t, t),
> $$
> where ${v}({x}\_t, t)=\mathbb{E}\_{p({x}\_0, {x}\_1\mid {x}\_t)}[\alpha\_t^\prime{x}\_0+\beta\_t^\prime{x}\_1]$ is the unconditional marginal velocity field, $c$ denotes conditions (e.g., class labels), ${v}({x}\_t,t\mid{c}) = \mathbb{E}\_{p({x}\_0, {x}\_1\mid {x}\_t, {c})}[\alpha\_t^\prime{x}\_0+\beta\_t^\prime{x}\_1]$ is the conditional marginal velocity field, and $w$ controls the CFG strength. Our conditional model thus satisfies the following two equations:
> $$
> \begin{align}
> &{f}\_{\theta}({x}\_t,t,t,c)={x}\_t,\\\\
> &\partial\_1{f}\_{\theta}({x}\_t,t,s,c){v}\_g({x}\_t,t,c)+\partial\_2{f}\_{\theta}({x}\_t,t,s,c)={0}.
> \end{align}
> $$
> This formulation shows that different conditions $c$ yield distinct denoising processes, necessitating the integration of $c$ into the model. Additional details can be found in our updated paper.

---

> ### Author Response · Authors · 2025-11-28
>
> > W2: I think the author should add a section to illustrate how to achieve distillation. Well, training from scratch is necessary, but distillation is also essential and can further enhance the contribution of this paper (can refer to the consistency models).
>
> While our paper primarily focuses on the training-from-scratch paradigm, we appreciate the suggestion to demonstrate how SoFlow applies to distillation. Extending our method to distillation is straightforward. In this setting, **the primary difference is that the mixed velocity field is derived from a pretrained teacher model rather than being predicted by the student model itself.** Specifically, in from-scratch training, the mixed velocity field is defined as:
> $$
> v\_{\text{mix}}=m(w(\alpha\_t^\prime x\_0+\beta\_t^\prime x\_1)+(1-w)v\_{\text{uncond}})+(1-m)v\_{\text{guide}},
> $$
> where $v\_{\text{uncond}}$ is predicted by $\partial\_2a(t,t)x\_t+\partial\_2b(t,t)F\_{\theta^-}(x\_t,t,t,\phi)$ and $v\_{\text{guide}}$ is predicted by $\partial\_2a(t,t)x\_t+\partial\_2b(t,t)F\_{\theta^-}(x\_t,t,t,c)$.
>
> In the distillation setting, we obtain the mixed velocity field from the teacher model. Since the teacher is trained to predict the conditional velocity field $v\_{\text{cond}}$ and the unconditional velocity field $v\_{\text{uncond}}$ separately (rather than the guided velocity field directly), we construct $v\_{\text{guide}}$ via $wv\_{\text{cond}}+(1-w)v\_{\text{uncond}}$. Substituting this linear combination into the original mixed velocity field formula, we derive the mixed velocity field for distillation:
> $$
> \begin{aligned}
> v\_{\text{mix}}&=m(w(\alpha\_t^\prime x\_0+\beta\_t^\prime x\_1)+(1-w)v\_{\text{uncond}})+(1-m)(wv\_{\text{cond}}+(1-w)v\_{\text{uncond}})\\\\
> &=w(m(\alpha\_t^\prime x\_0+\beta\_t^\prime x\_1)+(1-m)v\_{\text{cond}})+(1-w)v\_{\text{uncond}}.
> \end{aligned}
> $$
>
> It is worth noting that in the original formulation, the mix ratio $m$ cannot be set to $0$; doing so leads to a trivial solution where $v\_{\text{mix}}=v\_{\text{guide}}$, causing the flow matching loss to collapse to $0$ because $v\_{\text{guide}}$ is the model's own prediction. However, in the distillation setting, since $v\_{\text{mix}}$ is fully provided by the frozen teacher model, setting $m=0$ does not result in a collapsed loss for the student and is therefore a valid configuration.
>
> **To illustrate the differences in hyperparameter selection between distillation and training from scratch, we provide ablation study results using the DiT-B/4 architecture.** We trained a teacher diffusion model for 300K iterations, initialized the student model from the teacher, and then trained the student model for a further 100K iterations on the ImageNet-$256 \times 256$ dataset. We utilized a linear flow matching trajectory with Euler parameterization and set the CFG strength to 3.0, following the experimental setup in our paper.
>
> **Table: Ablation studies results for distillation situation**
> | (a) $l\to t$ schedule $r(k,K)$ | FID-50K (1-NFE) | (b) flow matching data ratio | FID-50K (1-NFE) |
> | :--- | :---: | :--- | :---: |
> | Exponential | **11.20** | 0% | **11.20** |
> | Cosine | 11.24 | 25% | 12.93 |
> | Linear | 11.22 | 50% | 14.31 |
> | Constant | 11.41 | 75% | 16.01 |
>
> | (c) loss weighting coefficient $p$ | FID-50K (1-NFE) | (d) velocity mix ratio $m$ | FID-50K (1-NFE) |
> | :--- | :---: | :--- | :---: |
> | 0.0 | 18.97 | 0.0 | **11.20** |
> | 0.5 | 12.14 | 0.25 | 11.84 |
> | 1.0 | **11.20** | 0.5 | 13.83 |
> | 1.5 | 12.61 | 0.75 | 15.59 |
>
> We summarize our findings from the distillation ablation studies as follows:
> *   (a) $l\to t$ **Schedule Function $r(k,K)$:** Choosing schedule functions $r(k,K)$ that decay from a higher value to a lower value (e.g., Exponential) yields better performance than a constant schedule, as this facilitates a smoother learning process.
> *   (b) **Flow Matching Data Ratio:** Unlike the training-from-scratch scenario, since the student model is initialized from a flow matching teacher, performance remains strong even when the flow matching data ratio is reduced to $0\%$. However, we hypothesize that for larger-scale models, a higher flow matching data ratio may be necessary to stabilize the training process.
> *   (c) **Loss Weighting Coefficient $p$:** Selecting an adaptive scaling function with medium robustness ($p=1$) remains beneficial in the distillation setting, which aligns with our findings for training from scratch.
> *   (d) **Velocity Mix Ratio $m$:** Since the teacher model already produces a high-quality estimation of the velocity field, removing the random term $\alpha\_t^\prime x\_0+ \beta\_t^\prime x\_1$ by setting the velocity mix ratio $m$ to $0$ leads to optimal performance in this setup. However, we note that choosing a non-zero $m$ might be advantageous for larger models trained for more steps, as $\alpha\_t^\prime x\_0+ \beta\_t^\prime x\_1$ contains ground-truth data information that may not have been learned by the teacher model.

---

> ### Author Response · Authors · 2025-11-28
>
> > W3 Experimental results in the Table. 1 has some inconsistency in the Table. 2. For example, the noise scheduler part shows the best FID is 59.97. But the true best FID is 3.35.
>
> We thank the reviewer for this observation. The apparent discrepancy arises because Table 1 and Table 2 report results from different experimental settings.
>
> **Table 1 presents ablation studies performed on the relatively compact SoFlow-B/4 model, designed to analyze component contributions. In contrast, Table 2 reports large-scale experiments using the significantly larger SoFlow-B/2, SoFlow-M/2, SoFlow-L/2, and SoFlow-XL/2 models.**
>
> The bolded best FID-50K scores in Table 1 (59.97 for unconditional and 12.89 for conditional generation) are therefore expected to be higher than the 3.35 FID-50K achieved by SoFlow-XL/2 in Table 2. This is consistent with the different purposes and model scales—Table 1 focuses on ablation analysis rather than achieving state-of-the-art performance. **Thus, there is no conflict between the reported values.**
>
> In addition, we find that decreasing the hyperparameter $r_{\text{end}}$ from $\frac{1}{320}$ to $\frac{1}{500}$ yields some performance improvements. We have rerun the ablation studies, and the results are provided in our updated paper. Currently, the best FID-50K scores are 58.57 for unconditional generation and 11.59 for conditional generation.
>
> > Q1 How much influence does using the expectation of the velocity to replacing the velocity term? Is there any way to quantify the bias after replacing it with expectation?
>
> When $l$ is close enough to $t$, replacing the velocity function $v(x\_t,t)=\mathbb{E}\_{p(x\_0,x\_1\|x\_t)}[\alpha\_t^\prime x\_0+\beta\_t^\prime x\_1]$ with $\alpha\_t^\prime x\_0+\beta\_t^\prime x\_1$ only introduces negligible approximation error. For simplicity, we consider minimizing the mean square error without the adaptive weighting function as $l\to t$:
>
> $$
> \begin{aligned}
>     &\min\_\theta\mathbb{E}[\||f\_\theta(x\_t,t,s)-f\_{\theta}(x\_t+v(x\_t,t)(l-t),l,s)\||\_2^2]\\\\
>     &=\min\_\theta \mathbb{E}[\||f\_\theta(x\_t,t,s)-f\_{\theta}(x\_t+\mathbb{E}[\alpha\_t^\prime x\_0+\beta\_t^\prime x\_1\|x\_t\](l-t),l,s)\||\_2^2]\\\\
>     &=\min\_\theta \mathbb{E}[\||(\partial\_1f\_\theta(x\_t,t,s)\mathbb{E}[\alpha\_t^\prime x\_0+\beta\_t^\prime x\_1\|x\_t]+\partial\_2 f\_\theta(x\_t,t,s))(t-l)+o(t-l)\||\_2^2]\\\\
>     &=(t-l)^2(\min\_\theta \mathbb{E}[\||\partial\_1f\_\theta(x\_t,t,s)\mathbb{E}[\alpha\_t^\prime x\_0+\beta\_t^\prime x\_1\|x\_t]+\partial\_2 f\_\theta(x\_t,t,s)\||\_2^2])+o((t-l)^2)\\\\
>     &=(t-l)^2(\min\_\theta \mathbb{E}[\||\partial\_1f\_\theta(x\_t,t,s)(\alpha\_t^\prime x\_0+\beta\_t^\prime x\_1)+\partial\_2 f\_\theta(x\_t,t,s)\||\_2^2])+o((t-l)^2),
> \end{aligned}
> $$
>
> where the final step applies the standard statistical property that the expectation minimizes the mean square error. We can convert this back to our objective:
>
> $$
> \begin{aligned}
>  &=\min\_\theta \mathbb{E}[\||(\partial\_1f\_\theta(x\_t,t,s)(\alpha\_t^\prime x\_0+\beta\_t^\prime x\_1)+\partial\_2 f\_\theta(x\_t,t,s))(t-l)+o(t-l)\||\_2^2]\\\\
>   &=\min\_\theta \mathbb{E}[\||f\_\theta(x\_t,t,s)-f\_{\theta}(x\_t+(\alpha\_t^\prime x\_0+\beta\_t^\prime x\_1)(l-t),l,s)+o(t-l)\||\_2^2]\\\\
>   &=\min\_\theta\mathbb{E}[\||f\_\theta(x\_t,t,s)-f\_{\theta}(x\_t+(\alpha\_t^\prime x\_0+\beta\_t^\prime x\_1)(l-t),l,s)\||\_2^2]+o((t-l)^2).
> \end{aligned}
> $$
>
> **During training, the scheduler $r(k,K)=\frac{l-t}{s-t}$ decays to a small value, ensuring that our objective ultimately becomes a high-precision approximation, which means that replacing the expectation with random estimation only introduces negligible discrepancy.**
>
> We sincerely thank you for your commendation of the project's potential and your valuable feedback, which has significantly enhanced our work. We hope these revisions address your concerns. Thank you again for your time and insights.

---

### Official Review · Reviewer_izTu · 2025-10-31

**Soundness:** 3
**Presentation:** 3
**Contribution:** 2
**Rating:** 6
**Confidence:** 5

**Summary:**

This manuscript proposes a variant of consistency models, which can be trained from scratch without JVP and achieve better 1-NFE FID than MeanFlow on ImageNet 256x256. The method involves a combination of flow matching and consistency training loss with carefully designed scheduling, loss weighting, and CFG handling. The results are promising, showing that JVP is not the only way to achieve comparable results to MeanFlow on ImageNet.

**Strengths:**

- The ImageNet 1-NFE results of the proposed method is very strong, beating MeanFlow without using the inefficient JVP operation. This could imply a very positive message to the community: JVP may not be really necessary for strong performance in consistency training/distillation. Rather, a well-designed training objective with careful CFG handling could also achieve something very competitive.
- All technical details (theories, design choices, ablation results) look very sound and the presentation quality is good. Notably, the introduction of velocity mix ratio is an interesting design choice, which clearly makes sense in terms of reducing variance and accelerating convergence.

**Weaknesses:**

- While I'm very positive on based on the evaluation results, this manuscript does not really introduce novel fundamental approaches that clearly distinguish it from prior art. For example:

  - The model architecture is similar to prior CM extensions that learns mappings from arbitrary t to s (e.g., CTM, MeanFlow)
  - Combining FM loss with consistency loss is also seen in prior work (e.g., CTM)
  - The CFG handling method can be broadly seen as an online guidance distillation approach (where the unconditional velocity is modeled by an unconditioned network prediction [1, 2]). MeanFlow also uses similar CFG handling (which is very important to FID).
  - For the consistency loss, the velocity mixing design is essentially mixing the consistency distillation objective (with $v_\text{guided}$ as the teacher, essentially it's online self distillation) with the consistency training objective (using real data).

  While this work does a very good job integrating all these techniques together, the limited novelty makes it difficult for a clear accept in my opinion.

- No multi-step results are presented in the paper. Multi-step generation is closer to applications of large-scale models. MeanFlow demonstrated very strong 2-NFE results. Beating it on 2-NFE could have further strengthened this work.

[1] Chen et al. Visual Generation Without Guidance. ICML 2025

[2] Tang et al. Diffusion Models without Classifier-free Guidance

**Questions:**

A few questions about design choices:
- Is a randomly sampled $s$ really necessary? For 1-NFE sampling, theoretically $s$ can just be zero. I wonder if there is a performance gain when using a random $s$ compared to setting $s=0$.
- Regarding the velocity mix ratio, $m=0$ is not tested in the ablation studies. Theoretically $m=0$ could work as long as there is a FM loss, basically the FM loss learns the velocity from data, and then the consistency loss at $m=0$ reduces to consistency distillation from the learned velocity (basically it's online self distillation).

---

> ### Author Response · Authors · 2025-11-28
>
> Thank you for the detailed and constructive feedback! We treasure the opportunity to address your concerns and improve our work.
>
> > W1: While I'm very positive on based on the evaluation results, this manuscript does not really introduce novel fundamental approaches that clearly distinguish it from prior art. For example:
> >
> > * The model architecture is similar to prior CM extensions that learns mappings from arbitrary $t$ to $s$ (e.g., CTM, MeanFlow)
> > * Combining FM loss with consistency loss is also seen in prior work (e.g., CTM)
> > * The CFG handling method can be broadly seen as an online guidance distillation approach (where the unconditional velocity is modeled by an unconditioned network prediction [1, 2]). MeanFlow also uses similar CFG handling (which is very important to FID).
> > * For the consistency loss, the velocity mixing design is essentially mixing the consistency distillation objective (with $v_{\text{guided}}$ as the teacher, essentially it's online self distillation) with the consistency training objective (using real data).
> >
> > While this work does a very good job integrating all these techniques together, the limited novelty makes it difficult for a clear accept in my opinion.
>
> Regarding the points raised, we wish to elucidate that **our model introduces novel contributions, particularly evidenced by the distinct training loss under Euler parameterization when compared to CTM and MeanFlow.** Under Euler parameterization,
>
> $$
> \begin{aligned}
>     &f_\theta(x_t,t,s)-f_{\theta^-}(x_t+v(x_t,t)(l-t),l,s) \\\\
> =&(x_t+(s-t)F_\theta(x_t,t,s)) - (x_t+v(x_t,t)(l-t)+(s-l)F_{\theta^-}(x_t+v(x_t,t)(l-t),l,s) \\\\
> =&(s-t)\left(F_\theta(x_t,t,s)-\left(\frac{l-t}{s-t}v(x_t,t)+\frac{s-l}{s-t}F_{\theta^-}(x_t+v(x_t,t)(l-t),l,s\right)\right) \\\\
> =&(s-t)\left(F_\theta(x_t,t,s)-\left(r(k,K)v(x_t,t)+(1-r(k,K))F_{\theta^-}(x_t+v(x_t,t)(l-t),l,s\right)\right),
> \end{aligned}
> $$
>
> where $r(k,K)$ is our schedule function that decreases monotonically from the beginning to the end of training. This derivation indicates that, ignoring weighting coefficients, our loss function is proportional to the mean square error:
>
> $$
> \mathbb{E}\_{t,l,s,x\_t}\left[\|\|F\_\theta(x\_t,t,s)-\left(r(k,K)v(x\_t,t)+(1-r(k,K))F\_{\theta^-}(x\_t+v(x\_t,t)(l-t),l,s)\right)\|\|\_2^2\right].
> $$
>
> In contrast, the objective of MeanFlow model is proportional to
>
> $$
> \mathbb{E}\_{t,s,x\_t}[\|\|F\_\theta(x\_t,t,s)-(v(x\_t,t)-(t-s)(\partial\_1 F\_{\theta^-}(x\_t,t,s) v(x\_t,t)+\partial\_2 F\_{\theta^-}(x\_t,t,s))\|\|\_2^2],
> $$
> where we omit the adaptive weighting coefficient as it does not affect gradient directions. As for CTM, their consistency loss is considerably less efficient compared to our solution consistency loss. More specifically, their loss can be equivalently expressed using our notation as:
>
> $$
> \mathbb{E}\_{t,s,x\_t}\left[\left\|\|f_{\theta^-}(f_\theta(x_t,t,s),s,0)-f_{\theta^-}(f_{\theta^-}(x\_t+v(x\_t,t)(t-\Delta t),t-\Delta t,s),s,0)\right\|\|\_2^2\right],
> $$
>
> where $x_t$ is defined by the PF-ODE trajectory of a SDE with exploding variance following Consistency Models, rather than taking a linear flow matching interpolation. Meanwhile, $f_\theta(x_t,t,s)$ is parameterized as $\frac{s}{t}x_t+(1-\frac{s}{t})F_\theta(x_t,t,s)$, rather than utilizing Euler parameterization.
> **The loss requires two additional forward propagations compared to our straightforward consistency loss.** Furthermore, CTM did not explore an effective method to approximate the loss and simply employs $\Delta t$ for different time points, whereas we introduce $r(k,K)=\frac{l-t}{s-t}$ to adaptively select $l$ for varying $s$ and $t$ during training. Lastly, we conduct experiments on effectively applying training-time CFG to our model, while CTM did not integrate CFG and instead relied on a GAN discriminator to enhance their model, which can introduce training instability. These distinctions render our model both different and more efficient relative to CTM.
>
> Evidently, under Euler parameterization, our model learns the average velocity function by linearly interpolating between the instantaneous velocity and the model's predicted average velocity at a lower noise level, with the interpolation coefficient $r(k,K)$ decreasing from $r_{\text{init}}$ to $r_{\text{end}}$ gradually during training process.
>
> **These equations clearly show that our solution consistency loss is not equivalent to MeanFlow loss and CTM loss.** Our schedule function $r(k,K)$, determined by the current training step and total steps, modulates the diffusive components in the target, enabling a smoother transition from flow matching models to solution flow models. By comparing the performance of exponential and constant schedules for $r(k,K)$, we further substantiate this idea.

---

> ### Author Response · Authors · 2025-11-28
>
> **Table 1: An ablation study on the larger SoFlow-B/2 model.**
>
> | Metrics | IS | FID | sFID | Precision | Recall |
> | :--- | :---: | :---: | :---: | :---: | :---: |
> | Constant $r(k,K)$ | 217.46 | 5.001 | 5.641 | 0.797 | 0.474 |
> | Exponential $r(k,K)$ | **218.85** | **4.849** | **5.356** | **0.801** | **0.477** |
>
> In our paper, ablation studies were conducted on SoFlow-B/4 models with 400K training steps to reduce computational costs, and the performance gap between the exponential and constant schedules was relatively small. Here, on the larger SoFlow-B/2 models trained for 240 epochs (1.2M training steps), the exponential schedule does improve performance, which we attribute to its smoother transition from flow matching to a solution flow model compared to the constant schedule.
>
> It is worth noting that the SoFlow-B/2 model here achieves a better FID-50K value than reported in the paper, due to adjusting the CFG strength to a more suitable value (see W2 for details).
>
> **In conclusion, our training target can be interpreted as linearly interpolating between instantaneous velocity fields and the model's predictions at lower noise levels, and is NOT merely integrating MeanFlow models, CTM and self-distillation techniques. This property allows our model to outperform MeanFlow when trained with the same architecture and steps, while eliminating the need for JVP computation.**
>
> We evaluated the training costs of MeanFlow and SoFlow using an identical PyTorch codebase, differing only by replacing our solution consistency loss with the MeanFlow loss. The results are presented below, which demonstrate that removing JVP computation is beneficial for improving training efficiency.
>
> **Table 2: Training Speed and GPU peak memory usage comparison.**
>
> | Methods | MeanFlow (Math Attn.) | Soflow (Math Attn.) | Soflow (Efficient Attn.) |
> | :--- | :---: | :---: | :---: |
> | GPU Memory | 51.45GB | 38.95 GB | 35.44GB |
> | Training Speed | 2.39 iters/sec | 2.84 iters/sec | 2.94 iters/sec |
>
> Currently, PyTorch's memory-efficient attention implementation supports standard forward and backward passes but lacks support for the Jacobian-Vector Product (JVP) computation required by MeanFlow. Consequently, MeanFlow must rely on the standard "math" attention backend. In contrast, our solution consistency loss is fully compatible with memory-efficient attention. As a result, **our model reduces peak GPU memory usage by approximately 31\% and increases training speed by 23\% compared to MeanFlow**, demonstrating significant gains in computational efficiency.

---

> ### Author Response · Authors · 2025-11-28
>
> > W2: No multi-step results are presented in the paper. Multi-step generation is closer to applications of large-scale models. MeanFlow demonstrated very strong 2-NFE results. Beating it on 2-NFE could have further strengthened this work.
>
> In response, we have conducted experiments on multi-step generation. To begin with, we find that the CFG strength of 3.0, previously determined via ablation on the smaller SoFlow-B/4 model, is **suboptimal** for our larger-scale models, which generally require lower guidance scales. By optimizing the CFG strength to 2.5, 2.25, 2.0, and 2.0 for SoFlow-B/2, M/2, L/2, and XL/2, respectively, we observe a **significant performance improvement**.
>
> With these optimized settings, SoFlow-XL/2 now achieves a 1-NFE FID-50K of **2.962** and a 2-NFE FID-50K of **2.661** after 240 training epoch, while MeanFlow-XL/2 reports a 1-NFE FID-50K of 3.43 and a 2-NFE FID-50K of 2.93. Furthermore, with tuned CFG scales, **our smaller models consistently surpass their larger MeanFlow counterparts**. Specifically, for 1-NFE generation, where MeanFlow reports FID-50K values of 5.01 (M), 3.84 (L), and 3.43 (XL), our optimized B, M, and L models achieve superior scores of 4.849, 3.733, and 3.201, respectively. These results demonstrate that our method not only delivers strong multi-step performance but also maintains a good trade-off between model size and performance.
>
> | Model | SoFlow-B/2 | SoFlow-M/2 | SoFlow-L/2 | SoFlow-XL/2 |
> | :--- | :---: | :---: | :---: | :---: |
> | CFG Strength | 2.5 | 2.25 | 2.0 | 2.0 |
> | Training Epochs | 240 | 240 | 240 | 240 |
> | FID-50K(1-NFE) | 4.849 | 3.733 | 3.201 | **2.962** |
> | FID-50K(2-NFE) | 4.244 | 3.423 | 2.899 | **2.661** |
>
>
> > Q1: Is a randomly sampled $s$ really necessary? For 1-NFE sampling, theoretically $s$ can just be zero. I wonder if there is a performance gain when using a random $s$ compared to setting $s=0$.
>
> While setting $s=0$ is theoretically permissible, we find that **using a randomly sampled $s$ yields better performance**. This is because our model learns the velocity field through a flow matching loss evaluated at $t=s$. **Fixing $s=0$ for the solution consistency loss creates a discontinuity between the two loss objectives**, which complicates the optimization process for the network.
>
> To validate this, we conducted an ablation study training SoFlow-B/4. When $s$ is fixed at $0$, the FID-50K score is $13.24$, which is worse than the score of $12.89$ achieved with a random $s$.
>
> > Q2: Regarding the velocity mix ratio, $m=0$ is not tested in the ablation studies. Theoretically $m=0$ could work as long as there is a FM loss, basically the FM loss learns the velocity from data, and then the consistency loss at $m=0$ reduces to consistency distillation from the learned velocity (basically it's online self distillation)
>
> We appreciate the reviewer's insight. However, in our from-scratch training scenario, **$m=0$ actually leads to training failure.** The mixed velocity field is defined as:
> $$
>     v\_{\text{mix}}=m(w(\alpha\_t^\prime x\_0+\beta\_t^\prime x\_1)+(1-w)v\_{\text{uncond}})+(1-m)v\_{\text{guide}}.
> $$
> This mixed velocity is used for computing both the Flow Matching loss and the solution consistency loss.
> **When $m=0$, the target for the Flow Matching loss becomes $v\_{\text{guide}}$, which is also the network's prediction. This causes the Flow Matching loss to vanish.** For the solution consistency loss, when $t$ is close to $s$, it exhibits similar behavior and its value approaches zero. This leads to a large adaptive weighting coefficient and results in severe training instability.
>
> We empirically tested a configuration where $m=0.25$ was used for the Flow Matching loss and $m=0$ for the solution consistency loss. Under our ablation settings, this yields an FID-50K of 35.62, which is significantly worse than the score of 12.89 achieved when using $m=0.25$ for both losses.
>
> In addition, we find that decreasing the hyperparameter $r_{\text{end}}$ from $\frac{1}{320}$ to $\frac{1}{500}$ yields some performance improvements. We have rerun the ablation studies, and the results are provided in our updated paper. Currently, the best FID-50K scores are 58.57 for unconditional generation and 11.59 for conditional generation. In the experiments for Q1 and Q2, we use the original $r_{\text{end}}$.

---

### Official Review · Reviewer_eADR · 2025-11-01

**Soundness:** 3
**Presentation:** 3
**Contribution:** 2
**Rating:** 6
**Confidence:** 4

**Summary:**

The paper derive a loss for learning the flow map $f(x_t, t,s)=x_s$ following the supervision of flow matching. The authors closely follow the derivation of Align Your Flow [1] and MeanFlow [2]. The main contribution of the paper is showing that applying the linear approximation of a small step (i.e., $t\rightarrow s$) as in equation 13 is not needed and the flow map can be learn with a finite difference approximation instead as in equation 14.

[1] Sabour, Amirmojtaba, Sanja Fidler, and Karsten Kreis. "Align Your Flow: Scaling Continuous-Time Flow Map Distillation." arXiv preprint arXiv:2506.14603 (2025).

[2] Geng, Zhengyang, et al. "Mean flows for one-step generative modeling." arXiv preprint arXiv:2505.13447 (2025).

**Strengths:**

1. The paper is well written (with slightly confusing notation).

2. The proposed method is backed by theory.

3. The proposed method achieve state-of-the-results (though comparable with baselines).

4. Proposed method suggest that in practice using JVP in [1,2] is not needed, potentially saving training cost.

**Weaknesses:**

1. the proposed method is of low novelty:

    a. The presented derivation is very close to Align Your Flow [1].

    b. The proposed preconditioning (referred as "Euler parametrization" ) of the flow map $f_{\theta}(x_t,t,s) = x_t + (s-t)F_{\theta}(x_t,t,s)$ was already suggested by [1] and essentially results in making the actual network parameters $F_{\theta}$ learn the mean field as in [1].

    c. Equation 13 was already used by [1], and composed with the Euler parametrization returns the mean flow objective [2].

    d. Approximating $v_t(x_t,t)=\mathbb{E}[\alpha_t' x_0 + \beta_t' x_1]$ with $\alpha_t' x_0 + \beta_t' x_1$ inside the flow map as in equation 14 is justified only for small $t-s$ where marginalization trick [3] can be applied as done in [2].

2. The statement that the proposed method "consistently outperforms Meanflow" (line 425) is somewhat limited given that in Table 2 for DiT-L and DiT-XL the difference between SoFlow and MeanFlow is 0.12 and 0.08 FID (resp.).

3. Though is seems the two method, SoFlow and MeanFlow, achieve comparable results, since SoFlow do not use JVP it can potentially reduce training cost which can be significant in large scale training. However, no empirical comparison or analysis of this is found in the paper.

4. No empirical validation of the method on the multistep setting. In particular the range 2-4 steps since previous work [1,2] reported state-of-the-art results in this range.

[3] Lipman, Yaron, et al. "Flow matching guide and code." arXiv preprint arXiv:2412.06264 (2024).

**Questions:**

Could the authors provide empirical comparison for training costs of SoFlow vs. MeanFlow ?

---

> ### Author Response · Authors · 2025-11-28
>
> We thank the reviewer for their thoughtful feedback and valuable suggestions. We provide detailed responses to all comments below.
>
> > W1: the proposed method is of low novelty:
> >
> > a. The presented derivation is very close to Align Your Flow [1].
> >
> > b. The proposed preconditioning (referred as "Euler parametrization" ) of the flow map $f\_\theta(x\_t,t,s)=x\_t+(s-t)F\_\theta(x\_t,t,s)$ was already suggested by [1] and essentially results in making the actual network parameters $F\_\theta$ learn the mean field as in [1].
> >
> > c. Equation 13 was already used by [1], and composed with the Euler parametrization returns the mean flow objective [2].
> >
> > d. Approximating $v(x\_t,t)=\mathbb{E}[\alpha\_t^\prime x\_0+\beta^\prime\_t x\_1]$ with $\alpha\_t^\prime x\_0+\beta^\prime\_t x\_1$ inside the flow map as in equation 14 is justified only for small where marginalization trick [3] can be applied as done in [2].
>
> Regarding points (a), (b), and (c), we wish to highlight the novelty of our model and clarify that, under Euler parameterization, **our training objective function can be regarded as a linear interpolation between the instantaneous velocity field and the model’s predicted average velocity field at lower noise levels. This formulation is fundamentally distinct from both MeanFlow [2] and Align Your Flow (AYF) [1].** Specifically, the difference term in our loss function can be expressed as follows:
>
> $$
> \begin{aligned}
>     &f\_\theta(x\_t,t,s)-f\_{\theta^-}(x\_t+v(x\_t,t)(l-t),l,s) \\\\
> =&(x\_t+(s-t)F\_\theta(x\_t,t,s)) - (x\_t+v(x\_t,t)(l-t)+(s-l)F\_{\theta^-}(x\_t+v(x\_t,t)(l-t),l,s)) \\\\
> =&(s-t)\left(F\_\theta(x\_t,t,s)-\left(\frac{l-t}{s-t}v(x\_t,t)+\frac{s-l}{s-t}F\_{\theta^-}(x\_t+v(x\_t,t)(l-t),l,s)\right)\right)\\\\
> =&(s-t)\left(F\_\theta(x\_t,t,s)-\left(r(k,K)v(x\_t,t)+(1-r(k,K))F\_{\theta^-}(x\_t+v(x\_t,t)(l-t),l,s)\right)\right),
> \end{aligned}
> $$
>
> where $r(k,K)$ is our schedule function, which decreases monotonically from the start of training to the end. Ignoring the weighting coefficients, this equation indicates that our loss function is proportional to the following mean squared error:
>
> $$
> \mathbb{E}\_{t,l,s,x\_t}\left[\|\|F\_\theta(x\_t,t,s)-\left(r(k,K)v(x\_t,t)+(1-r(k,K))F\_{\theta^-}(x\_t+v(x\_t,t)(l-t),l,s)\right)\|\|\_2^2\right].
> $$
>
> In contrast, the objective for the MeanFlow model is proportional to
>
> $$
> \mathbb{E}\_{t,s,x\_t}[\|\|F\_\theta(x\_t,t,s)-(v(x\_t,t)-(t-s)(\partial\_1 F\_{\theta^-}(x\_t,t,s) v(x\_t,t)+\partial\_2 F\_{\theta^-}(x\_t,t,s))\|\|\_2^2],
> $$
>
> where we omit the adaptive weighting coefficient as it does not affect the gradient direction. Regarding Align Your Flow [1], the authors demonstrate in Appendix D that under Euler parameterization, their objective is equivalent to MeanFlow [2].
>
> As shown above, under Euler parameterization, our model learns the average velocity function by linearly interpolating between the instantaneous velocity and the model's predicted average velocity at a lower noise level. Crucially, the interpolation coefficient $r(k,K)$ decreases from $r\_{\text{init}}$ to $r\_{\text{end}}$ during training, which gradually increases the approximation precision of our loss function.
>
> **These two equations clearly demonstrate that our solution consistency loss is not equivalent to AYF [1] or MeanFlow [2].** Our schedule function $r(k,K)$, determined by the current training step and total steps, controls the ratio of flow matching components in the target. This allows our model to transition smoothly from a flow matching model to a solution flow model. We demonstrate this by comparing the performance of an exponential schedule against a constant schedule $r(k,K)$.
>
> **Table 1: Ablation experiment on the larger SoFlow-B/2 model.**
>
> | Metrics | IS | FID | sFID | Precision | Recall |
> | :--- | :--- | :--- | :--- | :--- | :--- |
> | Const $r(k,K)$ | 217.46 | 5.001 | 5.641 | 0.797 | 0.474 |
> | Exponential $r(k,K)$ | **218.85** | **4.849** | **5.356** | **0.801** | **0.477** |
>
> While the ablation experiments in our paper were conducted on SoFlow-B/4 with 80 epochs to reduce computational costs (resulting in a relatively small performance gap), we present new results on the larger SoFlow-B/2 model trained for 240 epochs in Table 1. Here, the exponential schedule yields larger performance improvements. This confirms that the schedule provides a smoother transition strategy compared to a constant setting, distinguishing our method from prior work.
>
> (Note: The SoFlow-B/2 FID-50K value reported here improves upon the paper's results due to optimized CFG strength; see W2 for details.)
>
> **In conclusion, our training objective function can be regarded as a linear interpolation and is NOT simply an approximation of MeanFlow. This formulation allows our model to outperform MeanFlow (using the same architecture and training steps) while eliminating the JVP calculation.**

---

> ### Author Response · Authors · 2025-11-28
>
> Regarding (d), we clarify that **the approximation inside the flow map in Eq. 14 is justified when $\|l-t\|$ is small, rather than when $\|s-t\|$ is small**. For simplicity, consider minimizing the mean square error without the adaptive weighting function as $l\to t$:
>
> $$
> \begin{aligned}
>     &\min\_\theta\mathbb{E}[\|\|f\_\theta(x\_t,t,s)-f\_{\theta}(x\_t+v(x\_t,t)(l-t),l,s)\|\|\_2^2]\\\\
>     &=\min\_\theta \mathbb{E}[\|\|f\_\theta(x\_t,t,s)-f\_{\theta}(x\_t+\mathbb{E}[\alpha\_t\^\prime x\_0+\beta\_t\^\prime x\_1\|x\_t\](l-t),l,s)\|\|\_2^2]\\\\
>     &=\min\_\theta \mathbb{E}[\|\|(\partial\_1f\_\theta(x\_t,t,s)\mathbb{E}[\alpha\_t^\prime x\_0+\beta\_t^\prime x\_1\|x\_t]+\partial\_2 f\_\theta(x\_t,t,s))(t-l)+o(t-l)\|\|\_2^2]\\\\
>     &=(t-l)^2(\min\_\theta \mathbb{E}[\|\|\partial\_1f\_\theta(x\_t,t,s)\mathbb{E}[\alpha\_t^\prime x\_0+\beta\_t^\prime x\_1\|x\_t]+\partial\_2 f\_\theta(x\_t,t,s)\|\|\_2^2])+o((t-l)^2)\\\\
>     &=(t-l)^2(\min\_\theta \mathbb{E}[\|\|\partial\_1f\_\theta(x\_t,t,s)(\alpha\_t^\prime x\_0+\beta\_t^\prime x\_1)+\partial\_2 f\_\theta(x\_t,t,s)\|\|\_2^2])+o((t-l)^2),
> \end{aligned}
> $$
>
> where the final step applies the standard statistical property that the expectation minimizes the mean square error. We can convert this back to our objective:
>
> $$
> \begin{aligned}
>  &=\min\_\theta \mathbb{E}[\|\|(\partial\_1f\_\theta(x\_t,t,s)(\alpha\_t^\prime x\_0+\beta\_t^\prime x\_1)+\partial\_2 f\_\theta(x\_t,t,s))(t-l)+o(t-l)\|\|\_2^2]\\\\
>   &=\min\_\theta \mathbb{E}[\|\|f\_\theta(x\_t,t,s)-f\_{\theta}(x\_t+(\alpha\_t^\prime x\_0+\beta\_t^\prime x\_1)(l-t),l,s)+o(t-l)\|\|\_2^2]\\\\
>   &=\min\_\theta\mathbb{E}[\|\|f\_\theta(x\_t,t,s)-f\_{\theta}(x\_t+(\alpha\_t^\prime x\_0+\beta\_t^\prime x\_1)(l-t),l,s)\|\|\_2^2]+o((t-l)^2).
> \end{aligned}
> $$
>
> **During training, the scheduler $r(k,K)=\frac{l-t}{s-t}$ decays to a small value, ensuring that our objective ultimately becomes a high-precision approximation since $l$ is close to $t$. In conclusion, our approximation inside the flow map in Eq. 14 is valid.**

---

> ### Author Response · Authors · 2025-11-28
>
> > W2: The statement that the proposed method "consistently outperforms MeanFlow" (line 425) is somewhat limited given that in Table 2 for DiT-L and DiT-XL the difference between SoFlow and MeanFlow is 0.12 and 0.08 FID (resp.).
>
> **We attribute the previously observed marginal gap to suboptimal CFG strength.** Our new experiments reveal that the CFG strength of 3.0, which was determined via ablation studies on the smaller SoFlow-B/4 model, is excessive for larger models. Larger models generally require lower guidance scales; consequently, while SoFlow-B/2 and SoFlow-M/2 originally showed significant gains over MeanFlow, improvements for SoFlow-L/2 and SoFlow-XL/2 were limited by a large CFG strength.
>
> By reducing the CFG strength to 2.5, 2.25, 2.0, and 2.0 for SoFlow-B/2, M/2, L/2, and XL/2 respectively, **we achieved significant performance improvements**. With these optimized hyperparameters, SoFlow-XL/2 achieves a 1-NFE FID-50K of **2.962** and a 2-NFE FID-50K of **2.661** after 240 training epochs.
>
> Furthermore, with optimized CFG scales, **our models consistently outperform even larger MeanFlow counterparts**. Specifically, in the 1-NFE setting, MeanFlow models achieve FID-50K scores of 5.01 (M), 3.84 (L), and 3.43 (XL), whereas our optimized B, M, and L models achieve 4.849, 3.733, and 3.201, respectively.
>
> **Table 2: Optimized 1-NFE and 2-NFE results.**
>
> | Model | SoFlow-B/2 | SoFlow-M/2 | SoFlow-L/2 | SoFlow-XL/2 |
> | :--- | :---: | :---: | :---: | :---: |
> | CFG Strength | 2.5 | 2.25 | 2.0 | 2.0 |
> | Training Epochs | 240 | 240 | 240 | 240 |
> | FID-50K(1-NFE) | 4.849 | 3.733 | 3.201 | **2.962** |
> | FID-50K(2-NFE) | 4.244 | 3.423 | 2.899 | **2.661** |
> > W3: Though is seems the two method, SoFlow and MeanFlow, achieve comparable results, since SoFlow do not use JVP it can potentially reduce training cost which can be significant in large scale training. However, no empirical comparison or analysis of this is found in the paper.
>
> It is important to note that GPU memory consumption and training speed rely heavily on implementation details, hardware, and frameworks. To ensure a fair comparison, we evaluated the training costs of MeanFlow and SoFlow using an identical PyTorch codebase, differing only by replacing our solution consistency loss with the MeanFlow loss.
>
> The results are presented below. Currently, PyTorch's memory-efficient attention implementation supports standard forward and backward passes but lacks support for the Jacobian-Vector Product (JVP) computation required by MeanFlow. Consequently, MeanFlow must rely on the standard "math" attention backend. In contrast, our solution consistency loss is fully compatible with memory-efficient attention. As a result, **our model reduces peak GPU memory usage by approximately 31\% and increases training speed by 23\% compared to MeanFlow**, demonstrating significant gains in computational efficiency.
>
> **Table 3: Training Speed and GPU peak memory usage comparison.**
>
> | Methods | MeanFlow (Math Attn.) | Soflow (Math Attn.) | Soflow (Efficient Attn.) |
> | :--- | :---: | :---: | :---: |
> | GPU Memory | 51.45GB | 38.95 GB | 35.44GB |
> | Training Speed | 2.39 iters/sec | 2.84 iters/sec | 2.94 iters/sec |
>
> > W4: No empirical validation of the method on the multistep setting. In particular the range 2-4 steps since previous work [1,2] reported state-of-the-art results in this range.
>
> We address this concern with the results presented in Table 2 (refer to the response for W2). Specifically, **the SoFlow-XL/2 model achieves a 2-NFE FID-50K score of 2.661. This outperforms the MeanFlow-XL/2 baseline, which achieves 2.93** when trained from scratch for 240 epochs on ImageNet 256$\times$256. These results confirm the strong performance of our method in the multi-step (few-shot) regime.
>
> > Q1: Could the authors provide empirical comparison for training costs of SoFlow vs. MeanFlow ?
>
> We address this point in W3, where we empirically compare the training costs of SoFlow and MeanFlow.

---

### Author Response · Authors · 2025-12-04
**Summary for Rebuttal**

We are grateful to the reviewers for their constructive comments and suggestions, which have significantly helped us refine and strengthen our manuscript.

In summary, our rebuttal includes the following updates:

* **Clarification on Novelty (Reviewers 1 & 2):** Addressing concerns regarding the distinction between our method and prior works (e.g., MeanFlow, Align Your Flow, CTM), we have clarified that our learning objective differs significantly from these methods, thereby highlighting the novelty of our approach.

* **Performance on Large Models (Reviewer 1):** Regarding the observation that performance gains on larger models appeared smaller relative to baselines compared to smaller models, we identified that this was due to suboptimal hyperparameter selection. We have since optimized these parameters and provide stronger results in the revised manuscript.

* **Training Efficiency (Reviewer 1):** In response to concerns about training efficiency, we provide experimental results demonstrating that our model requires less GPU memory and achieves faster training speeds compared to the baseline model.

* **Approximation Error Analysis (Reviewers 1 & 3):** Addressing concerns about potential errors arising from approximating the solution function target (specifically, replacing the velocity field with a random term), we have provided a theoretical analysis proving that this error becomes negligible by the end of training.

* **Multi-step Evaluation (Reviewers 1, 2 & 4):** To address requests for multi-step evaluation, we have included competitive 2-NFE FID-50K results in the rebuttal.

* **Ablation on Hyperparameters (Reviewer 2):** Regarding the query on setting $m=0$ and $s=0$ during training, we demonstrate through experiments that these settings do not yield performance improvements.

* **Paper Presentation (Reviewer 3):** We have carefully revised the manuscript to address concerns regarding the writing and structure.

* **Distillation Capability (Reviewer 3):** In response to the question on the model's capability for distillation (in addition to our primary focus on training from scratch), we have provided further explanations and ablation studies.

* **Theoretical Guarantees (Reviewer 4):** Addressing theoretical concerns, we have provided a comprehensive analysis in W1 and W2. We show that our local training objectives ensure the model learns a global solution function and that our training loss implicitly minimizes the ODE error in both unconditional and conditional settings.

* **Comprehensive Metrics (Reviewer 4):** Following the request for a more comprehensive empirical evaluation, we have reported Inception Score (IS), sFID, Precision, and Recall metrics in addition to the original FID results.

* **Impact of Hyperparameters (Reviewer 4):** Regarding the request for clarification on the schedule function $r(k,K)$ and velocity mix ratio $m$, we have provided theoretical insights explaining how these terms influence empirical performance.

In conclusion, we have provided detailed responses to all concerns raised by the reviewers and have updated our paper accordingly. We believe our responses adequately address the reviewers' concerns.

---

### Meta-Review · Area_Chair_mg4u · 2026-01-08

**Summary:**

The paper proposes SoFlow, a variant of consistency models that eliminates the need for JVP while achieving improved one-step flow training. Reviewer scores are slightly biased towards acceptance (6, 6, 6, 4), reflecting appreciation for the sound approach, theoretical support, and strong experimental results. The authors made a notable effort to address the remaining concerns, significantly revising the manuscript including additional theoretical and empirical results. One concern from AC is that the extent of these changes is considerable so that the revision may require another full round of review to properly assess clarity and correctness. Nevertheless, AC believes that the merits of the paper outweigh this concern and supports acceptance.

**Reviewer Concerns:**

Overall, the reviewers raised the following major concerns:

- Limited technical novelty [eADR, izTu]
- Lack of multi-step evaluation [eADR, izTu, NydX]
- Lack of analysis on approximating the velocity term [eADR, aKZH]
- Presentation issues [aKZH]
- Theoretical concerns [NydX]

The rebuttal addresses these concerns in a substantive manner. In particular, the authors added multi-step evaluations, provided additional analysis, theorems, and discussion on the approximation of the velocity term, and improved the clarity and presentation of the manuscript.

**Reviewer Scores:**

- Reviewer eADR: Initially 6. Would maintain the original score if the novelty issue still stands
- Reviewer izTu: Initially 6. Would likely increase the score to above 6.
- Reviewer aKZH: Initially 6. Would maintain the original score if the novelty issue still stands
- Reviewer NydX: Initially 4. Would likely increase the score to 6.

---

### Decision · Program_Chairs · 2026-01-26

Accept (Poster)